# Comparative analysis of heparin affecting the biochemical properties of chicken and murine prion proteins

**Li-Juan Wang** [1,2]*, **Xiao-Dan Gu** [1,2], **Xiao-Xiao Li** [1,2], **Liang Shen** [1,2]*, **Hong-Fang Ji** [1,2]*

**1** Institute of Biomedical Research, Shandong University of Technology, Zibo, Shandong, People's Republic of China, **2** Shandong Provincial Research Center for Bioinformatic Engineering and Technique, Zibo Key Laboratory of New Drug Development of Neurodegenerative Diseases, School of Life Sciences, Shandong University of Technology, Zibo, Shandong, People's Republic of China

* wanglijuan@sdut.edu.cn (LJW); shen@sdut.edu.cn (LS); jhf@sdut.edu.cn (HFJ)

## Abstract

The conversion of cellular prion protein (PrP$^C$) to disease-provoking conformer (PrP$^{Sc}$) is crucial in the pathogenesis of prion diseases. Heparin has been shown to enhance mammalian prion protein misfolding. As spontaneous prion disease has not been reported in non-mammalian species, such as chicken, it is interesting to explore the influence of heparin on the conversion of chicken prion protein (ChPrP). Herein, we investigated the influences of heparin on biochemical properties of full-length recombinant ChPrP, with murine prion protein (MoPrP) as control. The results showed that at low heparin concentration (10 μg/mL), a great loss of solubility was observed for both MoPrP and ChPrP using solubility assays. In contrast, when the concentration of heparin was high (30 μg/mL), the solubility of MoPrP and ChPrP both decreased slightly. Using circular dichroism, PK digestion and transmission electron microscopy, significantly increased β-sheet content, PK resistance and size of aggregates were observed for MoPrP interacted with 30 μg/mL heparin, whereas 30 μg/mL heparin-treated ChPrP showed less PK resistance and slight increase of β-sheet structure. Therefore, heparin can induce conformational changes in both MoPrP and ChPrP and the biochemical properties of the aggregates induced by heparin could be modified by heparin concentration. These results highlight the importance of concentration of cofactors affecting PrP misfolding.

## Introduction

Incorrect folding of the mammalian normal prion proteins (PrP$^C$) into disease-provoking conformers (PrP$^{Sc}$) gives rise to a variety of neurodegenerative prion diseases, including 'mad cow disease' and human Creutzfeldt-Jakob disease [1, 2]. The infectious agent as suggested by protein-only hypothesis is composed of a misfold protein, PrP$^{Sc}$, and without nucleic acid [1]. Compared to the PrP$^C$, the PrP$^{Sc}$ has 1) different secondary structure, decreased α-helix (40% to 30%) and increased β-sheet (3% to 45%), 2) a great loss of solubility, 3) increased protease resistance and 4) aggregates in brain as amyloid structures [3, 4]. Despite all these studies,

gov.cn/ LJW National Natural Science Foundation of China (Grant Nos. 31500086) http://www.nsfc.gov.cn/ HFJ Shandong Provincial Natural Science Foundation (Grant Nos. JQ201508) http://cloud.sdstc.gov.cn/ LS Shandong Provincial Natural Science Foundation (Grant Nos. ZR2018MH010) http://cloud.sdstc.gov.cn/ HFJ Shandong Provincial Key Research and Development Program (Grant No. 2018GSF121001) http://cloud.sdstc.gov.cn/ The funders had no role in study design, data collection and analysis, decision to publish, or preparation of the manuscript.

**Competing interests:** The authors have declared that no competing interests exist.

neither the physiological role of PrP$^C$ nor the molecular mechanism of neurodegeneration in prion disease is clearly defined [5, 6]. The transmissibility of prion disease is widely regarded as that PrP$^{Sc}$ can act as a conformational template and interact with PrP$^C$ to create more PrP$^{Sc}$ [1, 7]. When propagating PrP$^{Sc}$ was implanted into mice with ablation of the PrP gene, no prion disease was observed in these host mice. This finding indicates that PrP$^C$ is critical and PrP$^{Sc}$ alone cannot cause prion disease [8–10].

Although PrP$^C$ is necessary for forming the transmissible PrP$^{Sc}$ in prion disease, efficient formation of the infectious agent is affected by interactions with cofactors, such as nucleic acids, lipids, glycosaminoglycans (GAGs), pH, and ionic characters [11–15]. Among these cofactors, GAGs are attractive since they were closely related to PrP$^{Sc}$ formation and PrP$^{Sc}$ deposits [16]. GAGs, especially heparan sulfate (HS), are found in amyloid deposits in prion disease or Alzheimer's disease, and HS is believed to be functionally involved in amyloid formation [16, 17]. Although almost no heparin, a hypersulfated analog of HS, is detected in the brain, heparin plays the same role of facilitating faithful replication of prions as the HS in protein misfolding cyclic amplification (PMCA) [18]. Moreover, heparin has been reported to promote the formation of β-sheet conformation in recombinant murine prion protein (MoPrP), leading to a protease-resistant form [19, 20]. Considering the commercial availability of heparin and the similarity with HS, heparin is used when many researchers investigated the interactions between PrP and GAGs [19, 21].

Aberrant structural changes of the PrP$^C$ to the infectious scrapie conformer PrP$^{Sc}$ cause prion diseases that affect a wide range of mammals. However, spontaneous prion disease was precluded by chickens [22, 23], and chickens challenged parenterally or orally with prion agent failed to be infected [23, 24]. The reason why chicken prion protein (ChPrP) is resistant to prion agent has aroused widespread concern [25–29]. Overall homology between mammalian and chicken prion proteins was low, the ChPrP amino acid sequence sharing 44% identity with that of the human [30]. However, all the essential features of mammalian PrP, such as signal peptide, tandem repeat domain, hydrophobic region and C-terminal globular domain, are observed in ChPrP [31]. In addition, the PrP expression profiles are similar in the central nervous system of mammals and chickens [32, 33]. Thus, there may be other factors associated with chicken's resistance to prion diseases [23].

Lots of elegant work has been done about the effects of GAGs on prion disease associated misfolding of mammalian PrP; however, little is known about the effect of GAGs on characteristics of ChPrP. Here, we investigated the ability of heparin to affect the biochemical properties of ChPrP and compared with the effect of heparin on MoPrP.

## Materials and methods

### Materials

Thioflavin T (ThT) and proteinase K (PK) were purchased from Sigma-Aldrich (T3516) and Merck, respectively. Heparin sodium salt from porcine intestinal mucosa (Millipore, 375095) was prepared in MilliQ water.

### Recombinant MoPrP and ChPrP expression and purification

According to previously reported protocol [34, 35], recombinant full-length PrP (MoPrP 23–230 and ChPrP 24–249) were expressed in *Escherichia coli* BL21 (DE3) and purified using Ni column followed by refolding. The Bradford assay was used to determine protein concentrations with BSA as a standard [36].

## Amyloid fibril formation assay

According to Abskharon *et. al* [37], amyloid fibril formation assays were carried out under denaturing conditions. PrP amyloid fibrils were generated by incubating the PrP (7.5 μM) in 100 mM potassium phosphate buffer containing 2 mol/L guanidinium chloride (GuHCl) and 20 μM thioflavin T (ThT) at pH 6.5. For investigating the effects of heparin, heparin was added to the reaction mixture at a final concentration of 45 μg/mL. The incubation was performed in a CLARIOstar (BMG) at 37°C and continuous shaking at 600 rpm. The progress of the reaction was monitored using a fluorometric ThT assay (excitation at 440 nm and emission at 480 nm).

## Thioflavin T assay

Samples consisted of 5 μM PrP, heparin (at the concentrations indicated) and 10 μM ThT were incubated at 25°C. The fluorescence was measured using a Varioskan FLASH (Thermo Scientific) according to Ellett et. al [19]. Excitation and emission wavelength were 430 and 480 nm, respectively.

## Protein solubility assay and proteinase K (PK) digestion

Protein solubility assay was carried out according to Ellett et. al [19]. Protein samples (5 μM) were incubated with 10 or 30 μg/mL of heparin at 25°C for 24 h. The pellet was separated by centrifugation (14,000 *g*, 10 min) and resuspended in the same volume of water, which was prepared for 12% acrylamide SDS-PAGE. To confirm the effects of high concentration of heparin, 30 μg/mL heparin were added to 5 μM PrP. After incubating for 24 h at 25°C, PK was used to digest the reaction mixtures at 37°C for 30 min with PK:PrP molar ratio of 1:480, 1:240, 1:120, 1:60, 1:30 and 1:15. The digestion was stopped by boiling for 10 min. Next 10 μL samples were subjected to 12% acrylamide SDS-PAGE.

## Circular dichroism (CD) and transmission electron microscopy (TEM)

These assays were performed according to a previously reported protocol [34]. Samples consisted of 5 μM PrP in solution or with 10 μg/mL or 30 μg/mL heparin were incubated at 25°C for 5 h. The pellet was separated by centrifugation (14,000 *g*, 30 min) and the supernatant was used for CD analysis. The ellipticity values (MilliQ water or heparin solution) were used as controls. For TEM, 4 μL incubating solution not centrifuged was fixed on 300 mesh copper grids (BZ10023b, Zhongjingkeyi), washed with 4 μL water, negatively stained using 2% uranyl acetate and examined on a Tecnai G2 Spirit TEM at voltage of 120 kV.

# Results

## The influence of heparin on capabilities of MoPrP and ChPrP to form amyloid fibrils

A previous study has shown that the capability of PrP to form amyloid fibrils is likely to be influenced by the environments of the protein [38]. To study the effects of heparin on fibril formation of PrP, amyloid fibril formation assays were conducted with MoPrP and ChPrP. Recombinant PrP (7.5 μM) treated with heparin (45 μg/mL) behaved similar to the PrP alone (Fig 1A). MoPrP shows a rapid growth phase in amyloid fibrils formation, while ChPrP has a longer lag phase and ThT fluorescence was much lower (Fig 1A). Whether the heparin is present or not, both recPrP species displayed similar lag times (Fig 1B) and maximum ThT

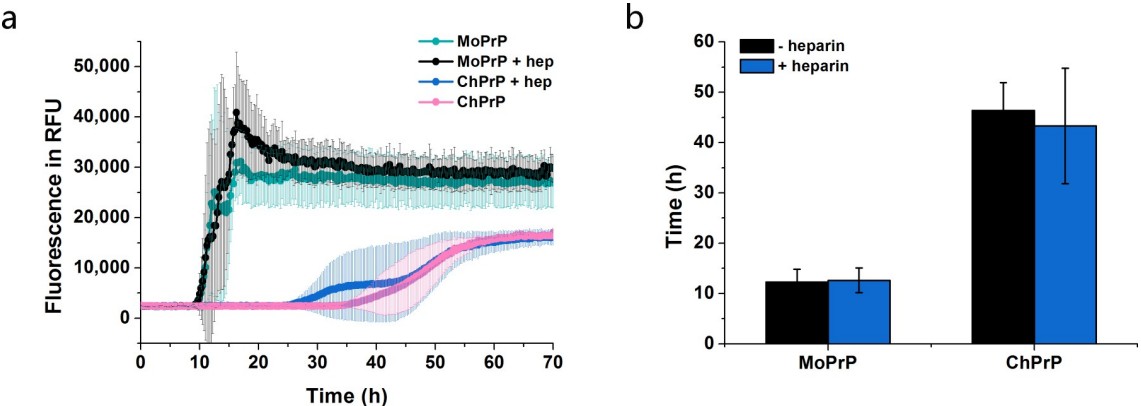

**Fig 1. Influences of heparin on amyloid fibril formation of PrP. a**, time-profile of PrP amyloid fibrils with or without heparin. PrP amyloid fibrils were generated by incubating the PrP (7.5 μM), ChPrP or MoPrP, in 100 mM potassium phosphate buffer containing 2 mol/L GuHCl and 20 μM ThT. **b**, influences of heparin on lag phase (time until the signal reached 10,000 RFU) of PrP. Error bars are the standard deviation (SD) of at least 3 repeats.

fluorescence intensity at plateau (Fig 1A), indicating that heparin may not play a crucial part in amyloid fibril formation under denaturing conditions.

## Thioflavin T fluorescence of heparin-treated ChPrP and MoPrP

GAGs involvement in the conversion of $PrP^C$ into $PrP^{Sc}$ under native conditions have been widely reported [18, 19]. To compare the effects of heparin on conformation transition of recombinant MoPrP (23–230) and ChPrP (24–249), we performed ThT fluorescence assays of PrP incubated with heparin in different concentrations. When either species of recPrP in MilliQ water was incubated with varied concentrations of heparin, little ThT reactivity was observed at low heparin concentrations (< 12.5 μg/mL), followed by a sharp fluorescence increase which starts at around 12.5 μg/mL heparin, and fluorescence saturation of PrP was reached at approximately 25 μg/mL heparin (Fig 2A). In addition, the saturated ThT

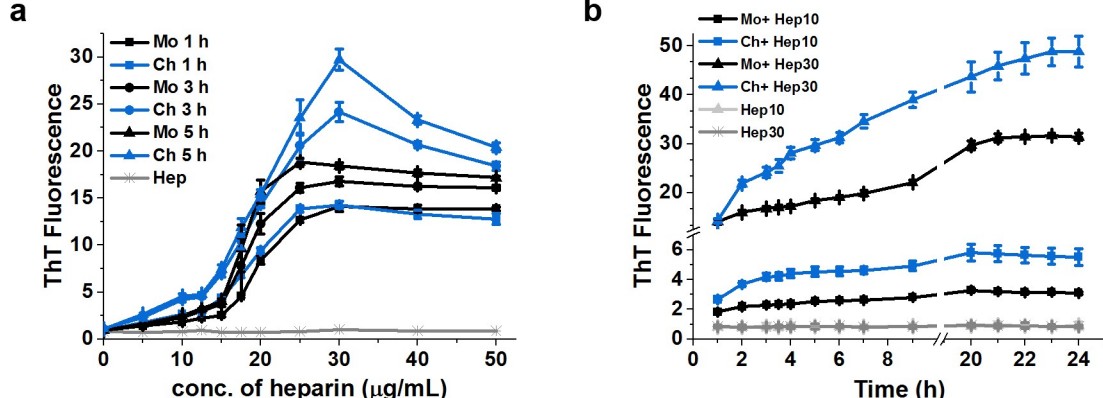

**Fig 2. Effects of heparin on ThT fluorescence of PrP in MilliQ water. a**, ThT fluorescences of PrP, 5 μM ChPrP (blue) or MoPrP (black), incubated with increasing concentrations of heparin were detected at different times. ThT fluorescences of increasing concentrations of heparin in MilliQ water (light gray) were measured as control. **b**, the time-profile of ThT fluorescence were detected using PrP (5 μM) after the addition of heparin at 10 μg/mL (square) or 30 μg/mL (triangle). Those of 10 μg/mL or 30 μg/mL heparin in solutions of MilliQ water alone (light gray and dark gray respectively) were also measured as controls. Error bars are the SD of at least 3 repeats.

fluorescences of MoPrP are more or less constant, while those of ChPrP increased gradually over time (Fig 2A). These results were confirmed when the interactions were performed in 10 mM sodium acetate buffer (pH 5.5) (S1A Fig). And PrP is more soluble in sodium acetate buffer than MilliQ water (S2 Fig).

Time-dependent changes in the ThT fluorescence of ChPrP and MoPrP were monitored in the presence of 10 and 30 μg/mL heparin (Fig 2B), before the sharp fluorescence increase occurred and after the fluorescence saturation respectively. When treated with 30 μg/mL heparin, the fluorescence values of MoPrP and ChPrP were significantly higher than those of PrP incubated with 10 μg/mL heparin. At the earliest time point, the ChPrP exhibited a similar fluorescence value compared to that of MoPrP regardless of whether the interactions were performed in the presence of 10 or 30 μg/mL heparin, whereas the fluorescence values of ChPrP grew faster than those of MoPrP. A similar situation with the same trend was observed when sodium acetate buffer was used as reaction solution (S1B Fig). The result suggests that PrP treated with 30 μg/mL heparin are more prone to form ThT positive aggregates than those treated with 10 μg/mL heparin. The pH of the MilliQ water was examined as 5~ 5.5 that was in keeping with the optimal pH of GAG-PrP interactions and was in keeping with the physiological pH at which prion conversion occurs [20, 38, 39]. To avoid the impact of buffers, subsequent experiments are carried out in the MilliQ water.

## Heparin alters the stability and proteinase K resistance of ChPrP and MoPrP

To investigate whether ChPrP and MoPrP became less soluble in the presence of heparin, the solubility of PrP treated with 10 or 30 μg/mL heparin was examined using centrifugation [19]. In the absence of heparin, a low intensity, full length MoPrP band was detected in pellet fraction after centrifugation, whereas heparin-treated MoPrP samples lost their solubility after the identical centrifugation step (Fig 3A). Moreover, 10 μg/mL heparin treatment of MoPrP

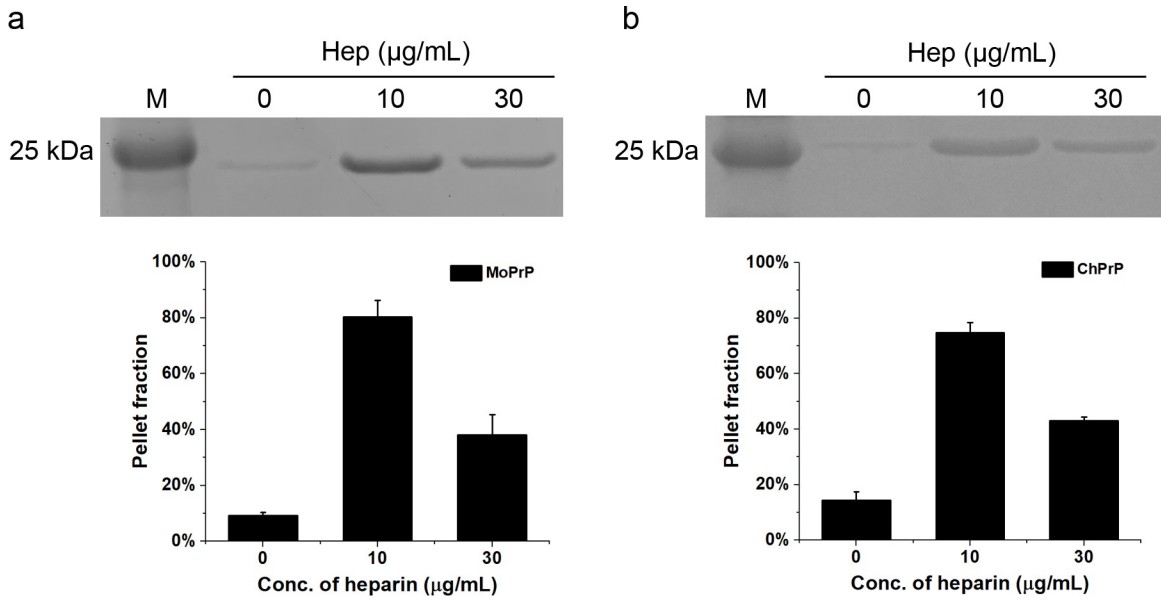

**Fig 3. Role of heparin treatment on the solubility of recPrP.** PrP at 5 μM, MoPrP (**a**) and ChPrP (**b**), were incubated with 10 or 30 μg/mL heparin for 24 h at 25˚C. The pellet was separated by centrifugation (14,000 *g*, 10 min) and resuspended in the same volume of water, which was prepared for SDS-PAGE. The fraction of PrP in pellet was quantified using ImageJ software and was shown as a percentage of total protein. The data are the average values of at least three replicates.

resulted in greater loss of solubility than that treated with 30 μg/mL heparin (Fig 3A), and a similar result was observed when ChPrP was used (Fig 3B). In the absence of heparin, weak band of ChPrP was observed in pellet fraction after centrifugation. Highly intense band of ChPrP were seen in pellet fraction after treated with 10 μg/mL heparin, whereas less intense band of ChPrP were detected when heparin concentration was 30 μg/mL (Fig 3B). The results above showed that PrP incubated with low concentration of heparin (10 μg/mL) were less soluble than those treated with heparin at high concentration (30 μg/mL).

Resistance to digestion with PK is one of the characteristics of PrP fibrils and PrP$^{Sc}$. As a result, we tested heparin-treated ChPrP and MoPrP for PK resistance. As almost all PrP treated with 10 μg/mL heparin were found in the pellet fraction after centrifugation (Fig 3), 30 μg/mL heparin was used when we analyzed the PK resistance. In the absence of heparin, the full-length MoPrP bands were only observed when the PK:PrP molar ratio was 1:480 and 1:240. After addition of 30 μg/mL heparin, the full-length MoPrP bands could be observed with PK:PrP molar ratio up to 1:60 (Fig 4A and 4B). In the absence of heparin, truncated ChPrP bands could be observed with PK:PrP molar ratio up to 1:30. However, after addition of 30 μg/mL heparin, no clear bands could be observed when PK:PrP molar ratio was 1:120 or greater (Fig 4C and 4D). Therefore, when the concentration of heparin was relatively high (30 μg/mL), heparin-treated MoPrP was more PK resistant whereas heparin-treated ChPrP was less PK resistant than PrP alone under the same condition.

## Structural changes in ChPrP and MoPrP treated with heparin

Secondary structures of ChPrP and MoPrP in the absence and presence of heparin were monitored using CD spectroscopy. The spectra of MoPrP and ChPrP in MilliQ water show the typical characteristic of a protein predominately consisted of α-helixes. The CD spectra of MoPrP and ChPrP almost changed to smooth curves after treated with 10 μg/mL heparin (Fig 5), which was in good agreement with the significant loss of solubility in PrP solubility assays (Fig 3). When MoPrP was treated with 30 μg/mL heparin, a dramatic decrease in the amplitude of the CD spectrum is observed (Fig 5A). In contrast, incubating of the ChPrP to 30 μg/mL heparin induces a relatively small decrease in the amplitude of the CD spectrum (Fig 5B).

## Morphology of MoPrP and ChPrP aggregates

The PrP aggregates were analyzed using TEM. Electron micrographs of negatively stained MoPrP and ChPrP in solution or incubated with 10 μg/mL heparin show no significant aggregates (Fig 6A, 6B, 6C and 6D). Compared to control samples that contain only MoPrP, ChPrP, or heparin (Fig 6A, 6B, 6E and 6H), many large spherical aggregates in the size up to 100 nm were observed when MoPrP was treated with 30 μg/mL heparin (Fig 6F). This suggests that MoPrP binds to heparin, undergoes conformational change and converts to large oligomers, which was in good agreement with previous studies [19]. These spherical aggregates were smaller than those reported in [38], with even more than 200 nm in size. This different was possibly due to using different heparin (30 μg/mL heparin v.s. 2 μM low molecular weight heparin) and incubating in different buffers (MilliQ water v.s. 10 mM acetate and 100 mM NaCl, pH 5.5). In contrast, a small number of dispersed aggregates with diameters less than 50 nm were observed for ChPrP with 30 μg/mL heparin (Fig 6G), which is consistent with a relatively small decrease in the amplitude in the CD spectrum of ChPrP treated with 30 μg/mL heparin (Fig 5B).

## Discussion

Prion diseases in many mammals are characterized by the PrP$^{Sc}$, a misfolded form of normal PrP$^{C}$ [1, 3]. The conversion of PrP$^{C}$ to PrP$^{Sc}$ is crucial in the pathogenesis of prion diseases

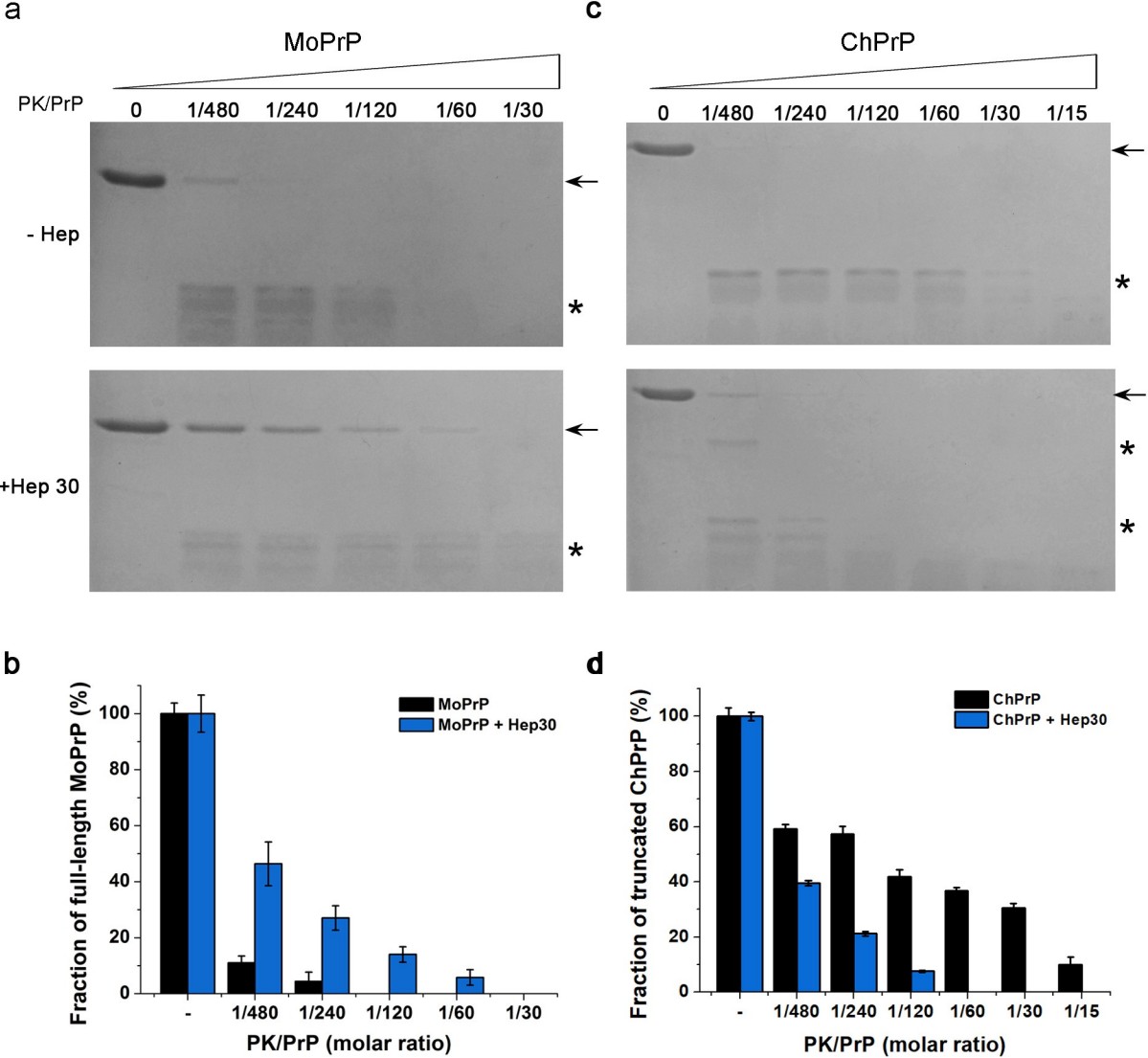

**Fig 4.** Effects of heparin on PK resistance of PrP. PrP at 5 μM, MoPrP (**a**) and ChPrP (**c**) in MilliQ water (-Hep) or treated with 30 μg/mL heparin (+Hep 30) were digested with PK for 30 min at 37°C. The PK/PrP molar ratio was 1:480, 1:240, 1:120, 1:60, 1:30 and 1:15 as indicated. Full-length PrP (→) and truncated digestion products (*) are indicated. **b**, the fraction of full-length MoPrP after PK digestion was quantified using ImageJ. The portion of full-length MoPrP is indicated as a percentage of total MoPrP protein without PK treatment. **d**, same as panel b but the portion of truncated ChPrP protein was indicated as a percentage of total ChPrP protein without PK treatment. The data are the average values of at least three replicates.

and heparin has been suggested to facilitate the conversion of mammalian PrP into protease-resistant forms [19, 20]. In contrast, prion diseases are excluded by non-mammals, such as chicken [22, 27]. In addition, little is known about how the characteristics of ChPrP can be affected by heparin. In the current study, the effects of heparin at different concentrations on the biochemical properties of full-length ChPrP were examined and compared with those of the biochemical properties of MoPrP. As far as we know, this is the first report about the effect of heparin on characteristics of ChPrP. Moreover, our results highlight the importance of cofactor concentrations that influence PrP misfolding.

Protein aggregation into amyloid fibrils is a pathological hall-mark of many diseases, including prion and Alzheimer's diseases [40, 41]. Here, we examined whether incubating

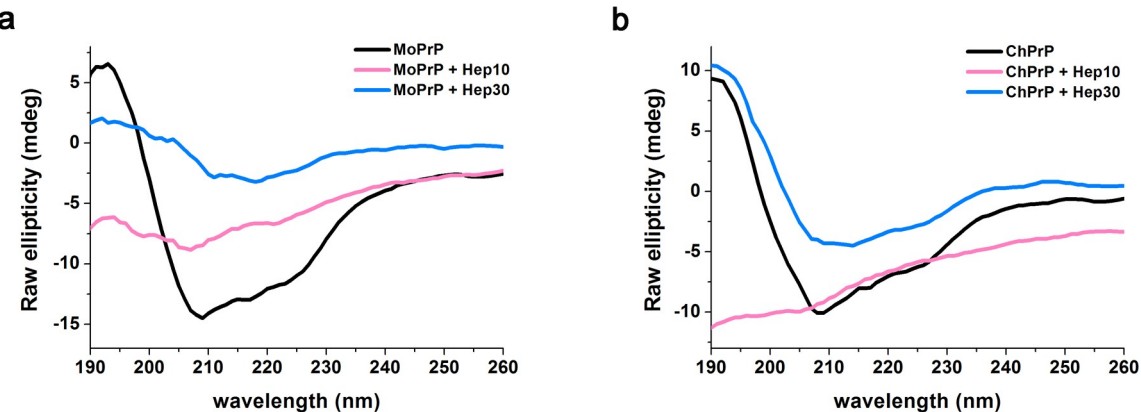

**Fig 5.** CD spectra of MoPrP (a) and ChPrP (b) in solution (black line) and after incubation with heparin at 10 μg/mL (pink line) or 30 μg/mL (blue line).

with heparin would affect the stability of PrP in amyloid fibril formation under denaturing conditions. Our result showed that heparin addition didn't play a role in amyloid fibril formation under denaturing conditions (Fig 1). However, low-molecular-weight heparin was shown to increase the stability of full-length PrP$^C$ from mice under denaturing conditions and seeded with infected brain tissue homogenate [42]. These could be explained by using different

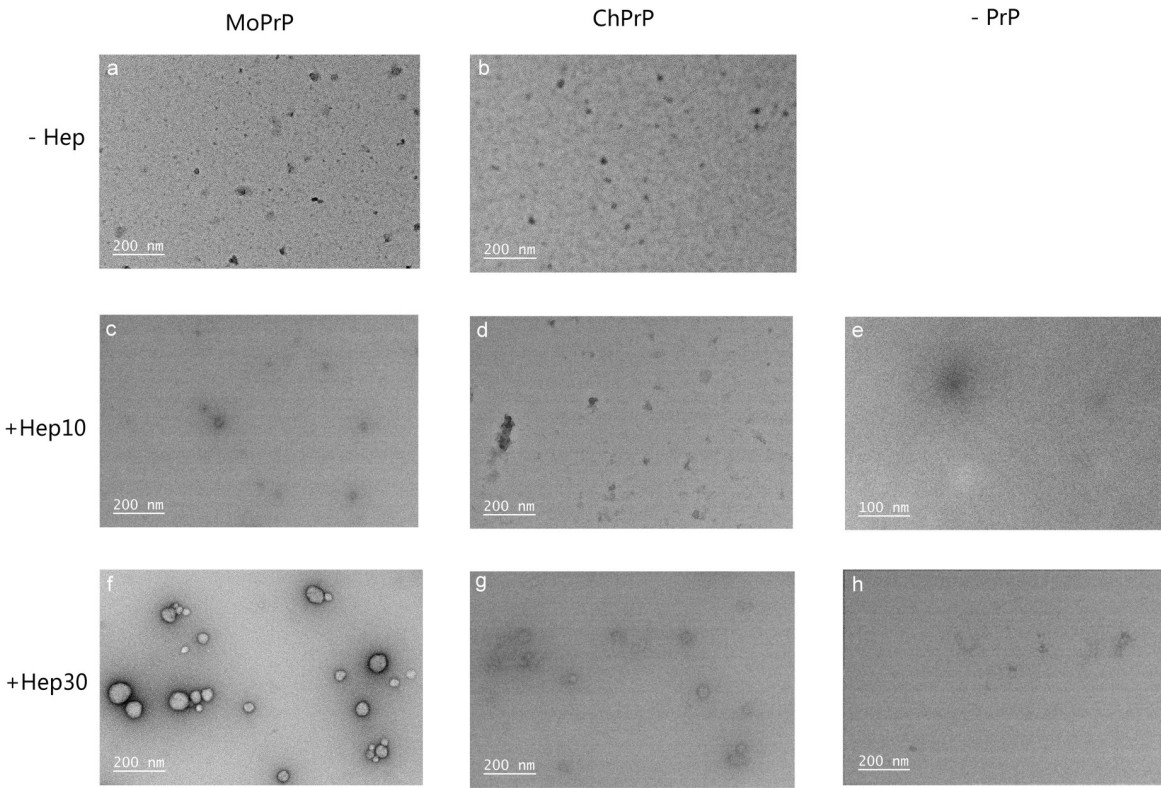

**Fig 6. Transmission electron micrographs of MoPrP and ChPrP.** PrP were incubated without (a and b) and with 10 μg/mL heparin (c and d) or 30 μg/mL heparin (f and g). There was no protein in 10 μg/mL heparin (e) or 30 μg/mL heparin (h). 5 μM PrP in water were incubated with or without heparin at 25°C for 5 h. A 2% uranyl acetate solution was used to negatively stain the samples.

heparin (30 μg/mL heparin v.s. 25 μM low molecular weight heparin) and incubating in different buffers (100 mM potassium phosphate buffer, 2 mol/L GuHCl, pH 6.5 v.s. 10 mM phosphate buffer, 130 mM NaCl, pH 7.4). Moreover, the influences of heparin on amplification efficiency of cell-PMCA, using recombinant human PrP as substrate, were seed dependent [15]. Thus, the conflicting results may also be explained by whether or which kinds of seeds were used in the reactions.

Heparin has previously been shown to induce conformational changes in mammalian PrP and play a part in the conversion of PrP^C to PrP^Sc [19, 20]. And yet, the function of sulfated GAGs is still debatable, since some studies suggested the opposite results, like that GAGs acts as protective factors preventing prion conversion [42, 43]. The present study confirmed that heparin can decrease MoPrP and ChPrP solubility and facilitate them to aggregate. Furthermore, our study revealed that biochemical properties of the aggregates differed depending on heparin concentrations. When 10 μg/mL heparin was used, a significant loss of solubility was observed for both MoPrP and ChPrP, whereas MoPrP and ChPrP interacted with high concentration heparin (30 μg/mL) were more soluble (Fig 3). This may be due to the ionic properties of heparin, as PrP in sodium acetate buffer pH 5.5 were more soluble than those in MilliQ water during the same centrifugation step (S2 Fig). Additionally, the electrostatic interaction of heparin with PrP might be modulated by salt concentration. In the current study, the reaction solutions with no NaCl might mask possible differences between affinity for MoPrP and ChPrP, which would be carried out in detail in our future studies. Moreover, the CD spectra of MoPrP and ChPrP treated with 10 μg/mL heparin almost became smooth curves (Fig 5). Far-UV CD spectroscopy is widely used as a valuable technique for analyzing structural changes of PrP, but protein aggregates can cause artifacts and distort the CD spectra [38, 44, 45]. Protein conformational changes might be masked by aggregates, even in the absence of protein precipitation [38]. Therefore, overall shift in CD spectra of MoPrP and ChPrP treated with 10 μg/mL heparin would be explained by the significantly decreased the solubility.

Heparin-treated MoPrP was more PK resistant, whereas the MoPrP was degraded by PK at concentrations lower than usually used for PrP fibrils or PrP^Sc. Here, PrP was incubated with heparin in MilliQ water without any seeds, which was not a suitable condition for fibril formation. Under this condition, the interaction between PrP and heparin causes widespread and persistent conformational changes to form an intermediate species of PrP^Sc [46]. These may be the reason why PK concentrations used here were lower than usually used for PrP fibrils or PrP^Sc. In contrast, ChPrP was more resistant to PK alone than ChPrP with 30 μg/mL heparin. The ChPrP treated with 30 μg/mL heparin showed the highest ThT fluorescence (Fig 2B), which suggests conformational changes were occurred after incubation with heparin. The formation of β-sheet conformation in prion protein usually leads to a protease-resistant form [19, 20]. However, PK-resistant form may not be the only destination of the conformational conversion [47]. The presence of 30 μg/mL heparin increased PK resistance and aggregate size of MoPrP (Figs 4 and 6), suggesting that high concentration of heparin induces a conformational change and contribute to the conversion of MoPrP to MoPrP^Sc. This finding is consistent with several studies showing that heparin directly influences the properties of PrP [19, 38]. As for ChPrP treated with 30 μg/mL heparin, less PK resistance and slight increase of β-sheet structure was observed. These findings are consistent with our previous study that full-length MoPrP and ChPrP interacted with the negatively charged lipid 1-palmitoyl-2-oleoyl-*sn*-glycero-3-phosphoglycerol (POPG) [34]. The presence of POPG increased β-sheet content, PK resistance and aggregate size of MoPrP, whereas POPG-treated ChPrP had decreased PK resistance and no obvious spherical aggregates [34]. The effects of high concentration of heparin (30 μg/mL) on characteristics of ChPrP were similar to those of POPG; however, the effects of these cofactors on characteristics of ChPrP differ from those on MoPrP. These results may

provide a new perspective on understanding the differences between mammalian and non-mammalian PrP and further on unraveling why prion diseases are only observed in mammals.

## Conclusions

In summary, our combined results of ThT fluorescence, solubility assay, PK resistance, CD and TEM show the effects of heparin on biochemical properties of MoPrP and ChPrP. Interaction with low concentration of heparin (10 μg/mL heparin) results in a significant loss of solubility for both MoPrP and ChPrP. High concentration of heparin (30 μg/mL heparin) has different influences on characteristics of MoPrP and ChPrP. Increased β-sheet content, PK resistance and size of aggregates were observed for MoPrP interacted with 30 μg/mL heparin, suggesting that heparin induces a conformational change and contribute to the conversion of PrP$^C$ to PrP$^{Sc}$. In contrast, 30 μg/mL heparin-treated ChPrP showed less PK resistance and slight increase of β-sheet structure. Therefore, the effects of heparin on the conformational changes of MoPrP and ChPrP varied in heparin concentration, which highlights the importance of concentration of cofactors affecting PrP misfolding. In addition, these results may provide a new perspective on understanding the differences between mammalian and non-mammalian PrP and further on unraveling why prion diseases are only observed in mammals.

## Supporting information

**S1 Fig. Effects of PrP-heparin interaction on ThT fluorescence in sodium acetate buffer (pH 5.5). a**, ThT fluorescences of PrP, 5 μM ChPrP (blue) or MoPrP (black), in the presence of increasing concentrations of heparin were detected at different times. ThT fluorescences of increasing concentrations of heparin in MilliQ water (light gray) were measured as control. **b**, the time-dependent changes in ThT fluorescence were determined for PrP (5 μM) after the addition of heparin at 10 μg/mL (square) or 30 μg/mL (triangle). Those of 10 μg/mL or 30 μg/mL heparin in solutions of MilliQ water alone (light gray and dark gray respectively) were also measured as controls. Error bars are the standard deviation (SD) of at least 3 repeats and are smaller than the symbol when absent in the figure.
(TIF)

**S2 Fig. The pellet of MoPrP and ChPrP in MilliQ water or in buffer pH 5.5.** The PrP at 5 μM was incubated in MilliQ water or in buffer pH 5.5 for 24 h at 25°C and then centrifugated for 10 min at 13,000 × g. Supernatant was separated from the pellet; the pellet was then resuspended in water to the same volume as the supernatant. The pellet samples were subjected to SDS-PAGE, visualized using Coomassie staining (**a**) and quantified using ImageJ software (**b**). The columns in panel b show the fold changes relative to the pellet portion of PrP in water. *P < 0.05 (Student's two tailed t-test), n = 3, mean ± SD.
(TIF)

**S1 Raw images.**
(TIF)

## Author Contributions

**Conceptualization:** Li-Juan Wang.

**Formal analysis:** Li-Juan Wang.

**Funding acquisition:** Li-Juan Wang, Liang Shen, Hong-Fang Ji.

**Investigation:** Li-Juan Wang, Xiao-Dan Gu, Xiao-Xiao Li.

**Methodology:** Li-Juan Wang, Xiao-Dan Gu, Xiao-Xiao Li.

**Project administration:** Liang Shen, Hong-Fang Ji.

**Supervision:** Liang Shen, Hong-Fang Ji.

**Validation:** Li-Juan Wang, Hong-Fang Ji.

**Writing – original draft:** Li-Juan Wang.

**Writing – review & editing:** Li-Juan Wang, Liang Shen, Hong-Fang Ji.

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
