## [Decision Letter · Decision Letter 0]

23 Nov 2020

PONE-D-20-18047

Comparative analysis of heparin affecting the biochemical properties of chicken and murine prion proteins

PLOS ONE

Dear Dr. Wang,

Thank you for submitting your manuscript to PLOS ONE. We also appreciate your extreme patience with our prolonged review process and for providing raw images of your gels (mistakingly called 'blots' in Reviewer 1's review below). We are now ready to invite you to address the reviewers' additional comments. As you can see below, both reviewers expressed interest in your study but also requested major revisions and clarifications. Having seen your raw images, my opinion is that you may ignore Reviewer 1's comment about the quality of Figure 4a + b. Otherwise, we look forward to a point-by-point response to the critiques.

We look forward to receiving your revised manuscript.

Kind regards,

Byron Caughey

Academic Editor

PLOS ONE

Journal Requirements:

Reviewers' comments:

Reviewer's Responses to Questions

**Comments to the Author**

1. Is the manuscript technically sound, and do the data support the conclusions?

Reviewer #1: Partly

Reviewer #2: Partly

2. Has the statistical analysis been performed appropriately and rigorously? 

Reviewer #1: N/A

Reviewer #2: N/A

3. Have the authors made all data underlying the findings in their manuscript fully available?

Reviewer #1: No

Reviewer #2: Yes

4. Is the manuscript presented in an intelligible fashion and written in standard English?

Reviewer #1: No

Reviewer #2: Yes

5. Review Comments to the Author

Reviewer #1: 1. Summary of the research and your overall impression

The authors biochemically and biophysically investigated the changes between a species of prion protein known to misfold into a disease associated form and a species not shown to be capable of this structural change when they are incubated with a potential cofactor mimetic.

Mouse PrP is known to be capable of undergoing conformational change to PrPSc and even recombinant MoPrP can instigate an infectious prion disease when combined with the right cofactors. This makes this protein a very sensible choice by the authors for this investigation. They also chose to use Chicken PrP as a non-disease-causing prion protein variant as prion diseases are mammalian diseases and are not known to affect birds. This enabled comparison between two protein homologues to investigate cofactor driven misfolding properties of PrP which make mammals victims of this disease.

Well established indicators of disease-associated misfolding were examined including fibril formation, ThT reactivity, solubility, Proteinase K resistance, increased beta-sheet secondary structure and aggregation. The addition of Heparin to MoPrP shifted the proteins properties to become more insoluble, aggregation prone, PK resistance and to possibly have a greater beta-sheet content when compared to untreated MoPrP, and ChPrP with and without Heparin treatment. MoPrP more readily forms ThT positive species under fibril generating conditions than ChPrP with Heparin not shown to contribute to this phenomenon.

The objective of this paper was to see if the prion protein from a class of animals susceptible to prion disease is more likely to develop disease associated biochemical and biophysical features both alone and in the presence of a potential cofactor than PrP from a class of animals resistant to prion disease and this objective was met but there are significant issues which need to be addressed. This paper also highlights an important property of prion protein behaviour in that it is very susceptible to concentration/molar ratio of cofactors. PrP behaviour can vary significantly with changes in treatment concentration. This review supports the publication of this data if the follow major issues are addressed adequately.

2. Discussion of specific areas for improvement

Line 51-58

The authors explain their use of Heparin as a mimetic for Heparan Sulfate by discussing its ability to generate prions in PMCA (ref 18) and its history of use as a mimetic. It is disappointing that the authors didn’t mention the findings of ref 19 that the lower levels of sulfation if Heparan Sulfate change the way PrP behaves biochemically when compared to Heparin. The authors will need to give an explanation of their decision to use Heparin in a way which is relevant to their specific study given that many experiments mirror those of ref 19.

Line 179-182

It is disappointing that the data showing the proportion of soluble to non-soluble PrP was not included. Can the authors state a reason for why they chose to include comparisons of pellet intensity (Fig 3) and not soluble vs insoluble PrP with and without Heparin treatment? Also, my understanding of agreeing to the statement “Yes - all data are fully available without restriction” implies you can’t have “data not shown”. This needs to be rectified

Line 215-218

The authors claim that the EM data for ChPrP treated with Heparin supports the small decrease in amplitude in the CD spectra. I don’t believe it is possible to draw such a conclusion without further experimental evidence. The size of aggregates under EM is unrelated to the secondary structure detected by CD.

Line 252-254

It is interesting to state that PrP is more soluble in sodium acetate buffer than water but inappropriate to bring new data into the Discussion section. S2 needs to be moved to the results section. The same comment applies as above - Can the authors state a reason for why they chose to include comparisons of pellet intensity and not soluble vs insoluble PrP? Can they also explain why they chose not to examine solubility during Heparin treatment in sodium acetate buffer?

Figure 3a + b

The blots are very poor quality and seem to be doctored. Can the authors provide raw, high quality files for these blots?

Figure 3c + d

y-axis should both have the same scale to allow easy comparison of MoPrP and ChPrP. Also, the y-axis label is insufficient and should mention that it is a fold increase over untreated pellet

Figure 4a + b

The blots are very poor quality. Can the authors provide raw, high quality files for these blots?

Figure 4 c + d

Can the authors explain why they chose two different populations of protein bands to quantify for these western blots? For this research question it would seem more appropriate to either quantify full length protein for both MoPrP and ChPrP or to quantify all bands present.

Figure 5a

MoPrP solubility with Heparin seems particularly poor in these researchers hands as the ellipticity is low and ragged at both Heparin concentrations. It is difficult to confidently claim Heparin induced beta-sheet structure in MoPrP from these results.

Figure 6 c and g

Images are very poor quality and seem out of focus. Can the authors provide higher quality, in focus images? It is also unusual that given how insoluble the PrP became upon addition of 10ug/ml of heparin that there were no aggregates seen in the EM images – can the authors provide a comment on these observations?

Grammatical changes:

Overall, the use of PrPs to describe the samples used should be changed to either PrP or recPrP – PrP does not need to be pluralised with an s.

Line 34

Change to “…protein, PrPSc, and without”

Line 35

Remove the word “obviously”

Line 36

Remove the word “the”

Line 37

Change to “aggregates”

Line 38

Change to “studies” not ‘researchers’

Line 41

Remove the word “However”

Line 43

Change from “only PrPSc” to “PrPSc alone”

Line 47

‘Characters’ is not the right word – try ‘molecules’ instead

Line 60

Change to “Causes prion diseases”

Line 71

Change to “…about the effects”

Line 129

Remove “the signal value of”

Line 130-131

Change to …or not, both recPrP species displayed…”

Line 133

Change to “under denaturing conditions”

Line 139-140

Change to “When either species of rec PrP in MilliQ…”

Line 158

Change to “form ThT positive aggregates”

Line 167

Change to …” a low intensity, full length MoPrP band was detected…”

Line 182

Remove “of PrPs”

Line 211

Change ‘much’ to ‘many’

Line 223-224

Change to “the conversion of mammalian PrP into protease-resistant forms”

Line 228

Change ‘that on’ to ‘those of’

Line 235

Change to ‘denaturing conditions’ and change ‘binding’ to ‘addition’

Line 266

Change to “directly influences the properties of PrP”

Line 500

Change to “Role of heparin treatment on the solubility of recPrP”

3. Any other points

I am happy to look over any revised manuscript

Reviewer #2: The article is original and compares the interaction of the glycosaminoglycan heparin with two sequences of PrP, murine and chicken. The effect of glycosaminoglycans, mainly heparin, on murine PrP protein was evaluated in different studies. However, the interaction with chicken PrP was unknown. The relevance of studying the effect on chicken PrP is mainly related to the fact that this animal does not develop prion diseases and, therefore, the differential interaction with this and other cofactors can bring important insight into this phenomenon.

The article shows the first description of the interaction between heparin and chicken PrP, demonstrating the effect of Hep on the oligomeric state of PrP. Importantly, the authors demonstrate a concentration-dependence for the observed effects, which is a very relevant finding. However, some results and interpretations will require clarification.

Main concerns:

- The experiments in figure 2 used ThT to follow PrP aggregation. However, the fiber morphology by TEM, and the CD signal are not characteristic of fibrillar forms. PrP:Hep 30 ug/mL samples showed the highest ThT fluorescence, but the smallest pellet. The authors should address and explain this contradiction. I suggest checking if the light scattering signal is not interfering with ThT signal. The authors should also show the ThT spectra of the samples with great variation to discard this doubt. The author could also use other approaches to confirm fibril formation.

- PK treatment was performed in order to show if the aggregates formed were resistant, “Resistance to digestion with PK is one of the characteristics of PrP fibrils and PrPSc” (line 178 and 179).

Figure 4 shows this data, but figures are not of good quality as protein is degraded by PK at concentrations lower than usually used for PrP fibrils or PrPSc. PrP:Hep 30 ug/mL was used, although it was less aggregated than PrP:Hep 10 ug/mL (Figure 3). ChPrP was more resistant to PK alone than MoPrP and both samples with Hep, although ChPrP:Hep 30 ug/mL showed the highest ThT fluorescence (Figure 1). Authors must clarify and discuss these contradictions. Also, authors should include the acrylamide concentration in the MM section.

- The authors used CD spectroscopy to show changes of PrP secondary structure. “When MoPrP was treated with 30 μg/mL heparin, a dramatic decrease in the amplitude of the CD spectrum is observed (Fig. 5a), the associated β-sheet content of which was 13.8-fold increase compared to MoPrP in solution. In contrast, incubating of the ChPrP to 30 μg/mL heparin induces a relatively small decrease in the amplitude of the CD spectrum (Fig. 5b), and only 1.3-fold increase in the β-sheet content was estimated relative to the secondary structure of ChPrP in solution” (lines 199-205).

The authors did not describe how they calculated the beta sheet content, and whether they used any spectrum deconvolution. A visual analysis of the result is not sufficient for this result. The data obtained with ChPrP + Hep 10 was not described. In addition, the MoPrP + Hep 30 data does not show enough ellipticity signal to carry out deconvolution. As the authors point out, the CD is an absorptive technique and changes in sample turbidity impair the analysis of secondary structure content. Another technique like FTIR would be more appropriate in this case. The dichroism data seems to corroborate with a greater aggregation and, therefore, a greater loss of signal, as observed in the TEM data of figure 6, but this does not allow inference about changes in the protein secondary structure. However, the data shows a different effect between murine and chicken proteins.

- “As for ChPrP treated with 30 μg/mL heparin, less PK resistance and slight increase of β-sheet structure was observed, suggesting that no significant conformational changes occurred and the infectious form PrPSc may not be the only destination of the conformational conversion [47, 48].” (lines 267-270). Conclusions like this must be reconsidered and rephrased.

Minor comments:

- Running title: I suggest changing for a more specific running title.

- All data were obtained in water or sodium acetate buffer, with no NaCl. The interaction of Hep with PrP is electrostatic, modulated by salt concentration. No NaCl may enhance the aggregation induced by Hep. Absence of NaCl can also mask possible differences between affinity for murine and chicken PrP. I suggest performing the experiment over different NaCl concentrations, comparing with the literature. Indeed, the effect showed to be different between the two PrP constructs.

- The authors centrifuged PrP:Hep samples and analysed the pellet at Figure 3. In the text its written “almost all PrPs treated with 10 μg/mL heparin were found in the pellet fraction after centrifugation (data not shown)” (line 180 and 181). I suggest showing the data, not only to compare the effect of different concentrations of Hep, but also to show how much of the total protein aggregates.

- MoPrP and ChPrP seems to be aggregated before Hep addition. Authors should consider this when analyzing all the data and interpreting the effect of heparin.

- Hep induced the formation of PrP spherical aggregates in reference 39 (supporting material), similar to what is being showed. The author should cite and compare.

- “Our result showed that heparin binding didn’t play a role in amyloid fibril formation under denaturation (Fig. 1). However, low-molecular-weight heparin was shown to increase the stability of full-length PrPC from mice under denaturation and seeded with infected brain tissue homogenate [43]” (lines 235-238). The authors should consider differences on sample preparation: PrP:Hep interaction was in denaturing buffer (this paper) and before addition to denaturing buffer (reference 43); Hep concentration and buffer are different.

- “The present study confirmed that heparin can decrease MoPrP and ChPrP stability and facilitate them to aggregate. Furthermore, our study revealed that biochemical properties of the aggregates differed depending on heparin concentrations.” (lines 247-250). The authors should be careful with this statement since inducing a decrease in solubility is not necessarily linked to a decrease in stability of the monomeric protein. Therefore, other experiments would be necessary to evaluate changes in stability.

6. PLOS authors have the option to publish the peer review history of their article (what does this mean?). If published, this will include your full peer review and any attached files.

Reviewer #1: **Yes: **Laura J Ellett

Reviewer #2: No

---

## [Author Response · Author response to Decision Letter 0]

3 Jan 2021

Dear Byron Caughey and Dear Reviewers:

Thank you for your efforts to handle and review our paper (PONE-D-20-18047) entitled: “Comparative analysis of heparin affecting the biochemical properties of chicken and murine prion proteins”. We have carefully revised the manuscript based on your suggestions. Followings are Listed Responses to the Editor and Reviewers’ Comments and Suggestions:

Reviewer 1:

Line 51-58

The authors explain their use of Heparin as a mimetic for Heparan Sulfate by discussing its ability to generate prions in PMCA (ref 18) and its history of use as a mimetic. It is disappointing that the authors didn’t mention the findings of ref 19 that the lower levels of sulfation if Heparan Sulfate change the way PrP behaves biochemically when compared to Heparin. The authors will need to give an explanation of their decision to use Heparin in a way which is relevant to their specific study given that many experiments mirror those of ref 19.

Responses: Thanks for your good suggestions. We have added the findings of ref 19 to the revised manuscript at Page 7 Lines 124-127. The revised parts were as followings: GAGs, especially heparan sulfate (HS), are found in amyloid deposits in prion disease or Alzheimer’s disease, and HS is believed to be functionally involved in amyloid formation [16, 17]. Although almost no heparin, a hypersulfated analog of HS, is detected in the brain, heparin plays the same role of facilitating faithful replication of prions as the HS in protein misfolding cyclic amplification (PMCA) [18]. Under physiological conditions, aggregates formed in the presence of heparin behaved differently in solubility and protease resistance when compared to those formed in the presence of HS. Nevertheless, both heparin and HS can promote the formation of β-sheet conformation in recombinant murine prion protein (MoPrP) [19]. Considering the commercial availability of heparin and the similarity with HS, heparin is used when many researchers investigated the interactions between PrP and GAGs.

In this study, it was intended to compare the different effects of GAGs on ChPrP and MoPrP. Considering the commercial availability of heparin and the similarity with HS, we used heparin. Meanwhile, HS is also very suitable for studying the influence of GAGs on PrP, and we would use HS in our subsequent studies. Thank again for your good suggestions.

Line 179-182

It is disappointing that the data showing the non-soluble PrP was not included. Can the authors state a reason for why they chose to include comparisons of pellet intensity (Fig 3) and not soluble vs insoluble PrP with and without Heparin treatment? Also, my understanding of agreeing to the statement “Yes - all data are fully available without restriction” implies you can’t have “data not shown”. This needs to be rectified

Responses: Thanks for your good suggestions and sorry for not showing the proportion of soluble to non-soluble PrP. During protein solubility assays, the same concentration of protein (5 μM) was used. After incubation and centrifugation, the protein was divided into two parts: pellet and supernatant. Since the total amount of protein is the same, the amount of protein in the supernatant is negatively correlated with the amount of protein in the pellet. Thus, we examined the content of pellet part using SDS-PAGE. Additionally, we revised Fig 3 to show pellet fraction of total PrP (5 μM) as the reviewer suggested. The revised Fig 3 could also support “almost all PrPs treated with 10 μg/mL heparin were found in the pellet fraction after centrifugation” and we deleted “data not shown” in the revised manuscript at Page 11 Line 217 to Page 12 Line 220. The newly revised Fig 3 was as followings:

Line 215-218

The authors claim that the EM data for ChPrP treated with Heparin supports the small decrease in amplitude in the CD spectra. I don’t believe it is possible to draw such a conclusion without further experimental evidence. The size of aggregates under EM is unrelated to the secondary structure detected by CD.

Responses: Thanks for your good suggestions. It is really true as the reviewer suggested that the size of aggregates under EM is unrelated to the secondary structure detected by CD. Negative staining TEM was used to determine the appearance of the β-sheet rich, ThT binding structures (Ellett LJ, et al. Glycobiology, 2015, doi: 10.1093/glycob/cwv014). A small number of dispersed aggregates with diameters less than 50 nm were observed for ChPrP with 30 μg/mL heparin (Fig 6g), which suggests a small amount of β-sheet rich structure formation. It may support the small decrease in amplitude in the CD spectra. And we revised the related sentences in the revised manuscript on Page 14 Lines 269-272. The revised sentences were as followings: In contrast, a small number of dispersed aggregates with diameters less than 50 nm were observed for ChPrP with 30 μg/mL heparin (Fig 6g), which suggests a small amount of β-sheet rich structure formation and is consistent with a relatively small decrease in the amplitude in the CD spectrum of ChPrP treated with 30 μg/mL heparin (Fig 5b). 

Line 252-254

It is interesting to state that PrP is more soluble in sodium acetate buffer than water but inappropriate to bring new data into the Discussion section. S2 needs to be moved to the results section. The same comment applies as above - Can the authors state a reason for why they chose to include comparisons of pellet intensity and not soluble vs insoluble PrP? Can they also explain why they chose not to examine solubility during Heparin treatment in sodium acetate buffer?

Responses: Thanks for your good suggestions. We have moved the S2 Fig to the Results section as the reviewer suggested. More details had been added to the revised manuscript at Page 9 Lines 163-164. The revised parts were as followings: PrP is more soluble in sodium acetate buffer than MilliQ water (S2 Fig). During protein solubility assays, the same concentration of protein (5 μM) was used. After incubation and centrifugation, the protein was divided into two parts: the pellet and the supernatant. Since the total amount of protein is the same, the amount of protein in the supernatant is negatively correlated with the amount of protein in the pellet. Thus, we examined the content of pellet part using SDS-PAGE. In our current study, Thioflavin T fluorescence of PrP was measured both in MilliQ water and sodium acetate buffer. The results were consistent in both solutions. To avoid the impact of buffers, subsequent experiments are carried out in the MilliQ water. Indeed, the effect was showed to be different between the two PrP constructs.Thanks very much for the reviewer’s good suggestion, heparin treatment in sodium acetate buffer is also a good choice, which would be carried out in our future studies.

Figure 3a + b

The blots are very poor quality and seem to be doctored. Can the authors provide raw, high quality files for these blots?

Response: Thank you for your careful reading. We have provided the original uncropped and unadjusted images underlying all gel results in supporting information.

Figure 3c + d

y-axis should both have the same scale to allow easy comparison of MoPrP and ChPrP. Also, the y-axis label is insufficient and should mention that it is a fold increase over untreated pellet

Responses: Thanks for your good suggestions. We have revised Fig 3 to show pellet fraction of total PrP (5 μM) as the reviewer suggested. The y-axis in the revised Fig 3 showed the same scale and the y-axis label was revised to “Pellet fraction”. The revised Fig 3 was as followings:

Figure 4a + b

The blots are very poor quality. Can the authors provide raw, high quality files for these blots?

Response: Thank you for your careful reading. We have provided the original uncropped and unadjusted images underlying all gel results in supporting information.

Figure 4 c + d

Can the authors explain why they chose two different populations of protein bands to quantify for these western blots? For this research question it would seem more appropriate to either quantify full length protein for both MoPrP and ChPrP or to quantify all bands present.

Response: Thank you for your good suggestions. When PK resistance was tested using MoPrP in the absence or presence of heparin, the major difference was the full-length MoPrP bands. Thus, we quantified full-length protein bands. While using ChPrP for PK resistance assay, we could only observe the full-length protein at heparin-treated ChPrP with PK:PrP molar ratio of 1:480. The major difference was the truncated ChPrP bands, thus we quantified the truncated protein bands. PK resistance of heparin-treated MoPrP and ChPrP was compared to MoPrP and ChPrP alone, respectively. Therefore, we quantify two different populations of protein bands to better demonstrate the differences.

Figure 5a

MoPrP solubility with Heparin seems particularly poor in these researchers hands as the ellipticity is low and ragged at both Heparin concentrations. It is difficult to confidently claim Heparin induced beta-sheet structure in MoPrP from these results.

Response: Thank you for your good suggestions. Indeed, the MoPrP + Hep30 data does not show enough ellipticity signal to calculate the β-sheet content, and it cannot be concluded from these CD spectra that heparin induced β-sheet structure in MoPrP. Thus, we revised the related sentences in the Results and Discussion sections. Details can be found in the revised manuscript at Page 13 Lines 247-248 and Page 17 Lines 325-327.

Figure 6 c and g

Images are very poor quality and seem out of focus. Can the authors provide higher quality, in focus images? It is also unusual that given how insoluble the PrP became upon addition of 10ug/ml of heparin that there were no aggregates seen in the EM images – can the authors provide a comment on these observations?

Response: Thank you for your good suggestions. Sorry for not providing high quality and focus images. No significant aggregates were found in Figs 6a, 6b, 6c, 6d, 6e and 6h, thus these pictures are not well focused. In contrast, the pictures of MoPrP and ChPrP treated with 30 μg/mL heparin are of higher quality. Indeed, PrP became insoluble upon addition of 10 μg/ml of heparin, whereas the protein in the pellet did not have to be aggregated. Electron micrographs of resuspended pellet showed no significant aggregates and the presence of the pellet did not imply conformational changes.

Grammatical changes:

1) Overall, the use of PrPs to describe the samples used should be changed to either PrP or recPrP – PrP does not need to be pluralised with an s.

2) Line 34

Change to “…protein, PrPSc, and without”

3) Line 35

Remove the word “obviously”

4) Line 36

Remove the word “the”

5) Line 37

Change to “aggregates”

6) Line 38

Change to “studies” not ‘researchers’

7) Line 41

Remove the word “However”

8) Line 43

Change from “only PrPSc” to “PrPSc alone”

9) Line 47

‘Characters’ is not the right word – try ‘molecules’ instead

10) Line 60

Change to “Causes prion diseases”

11) Line 71

Change to “…about the effects”

12) Line 129

Remove “the signal value of”

13) Line 130-131

Change to …or not, both recPrP species displayed…”

14) Line 133

Change to “under denaturing conditions”

15) Line 139-140

Change to “When either species of rec PrP in MilliQ…”

16) Line 158

Change to “form ThT positive aggregates”

17) Line 167

Change to …” a low intensity, full length MoPrP band was detected…”

18) Line 182

Remove “of PrPs”

19) Line 211

Change ‘much’ to ‘many’

20) Line 223-224

Change to “the conversion of mammalian PrP into protease-resistant forms”

21) Line 228

Change ‘that on’ to ‘those of’

22) Line 235

Change to ‘denaturing conditions’ and change ‘binding’ to ‘addition’

23) Line 266

Change to “directly influences the properties of PrP”

24) Line 500

Change to “Role of heparin treatment on the solubility of recPrP”

Responses: Thanks for your good suggestions. We have revised all these typos/grammatical errors.

Thanks to Reviewer 1 for your time and efforts to help us improve the quality of this MS.

Reviewer 2:

Main concerns:

- The experiments in figure 2 used ThT to follow PrP aggregation. However, the fiber morphology by TEM, and the CD signal are not characteristic of fibrillar forms. PrP:Hep 30 ug/mL samples showed the highest ThT fluorescence, but the smallest pellet. The authors should address and explain this contradiction. I suggest checking if the light scattering signal is not interfering with ThT signal. The authors should also show the ThT spectra of the samples with great variation to discard this doubt. The author could also use other approaches to confirm fibril formation.

Response: Thank you for your good suggestions. In figure 1, we studied the effects of heparin on fibril formation of PrP and found that heparin may not play a crucial part in amyloid fibril formation under denaturing conditions. In the rest of the manuscript, we did not study the formation and characteristics of fibers. The CD and TEM were used to monitor the conformational changes and aggregates of PrP. Indeed, PrP:Hep 30 μg/mL samples showed the highest ThT fluorescence, but the smallest pellet. However, the protein in the pellet did not have to be aggregated and electron micrographs of resuspended pellet showed no significant aggregates. Protein aggregation into amyloid fibrils is a pathological hall-mark of many diseases, including prion. We would investigate fibril formation in detail in our future studies. Thanks very much for the reviewer’s good suggestion again.

- PK treatment was performed in order to show if the aggregates formed were resistant, “Resistance to digestion with PK is one of the characteristics of PrP fibrils and PrPSc” (line 178 and 179).

Figure 4 shows this data, but figures are not of good quality as protein is degraded by PK at concentrations lower than usually used for PrP fibrils or PrPSc. PrP:Hep 30 ug/mL was used, although it was less aggregated than PrP:Hep 10 ug/mL (Figure 3). ChPrP was more resistant to PK alone than MoPrP and both samples with Hep, although ChPrP:Hep 30 ug/mL showed the highest ThT fluorescence (Figure 1). Authors must clarify and discuss these contradictions. Also, authors should include the acrylamide concentration in the MM section.

Response: Thank you for your good suggestions. In Fig 4, PrP was incubated with heparin in MilliQ water without any seeds, which was not a suitable condition for fibril formation. Under this condition, the interaction between PrP and cofactor causes widespread and persistent conformational changes to form an intermediate species of PrPSc (A. Zurawel, et al. Biochemistry, 2014, doi: 10.1021/bi4014825). These may be the reason why PK concentrations used in Fig 4 were lower than usually used for PrP fibrils or PrPSc. Indeed, PrP incubated with low concentration of heparin (10 μg/mL) were less soluble than those treated with heparin at high concentration (30 μg/mL) (Fig 3). However, the protein in the pellet did not have to be aggregated. The presence of pellet did not imply conformational changes, and most of the protein in the pellet is not conducive to subsequent experiments. Thus, 30 μg/mL heparin was used when we analyzed the PK resistance. ChPrP:Hep 30 μg/mL showed the highest ThT fluorescence (Fig 1), which suggests conformational changes were occurred after incubation with heparin. However, PK-resistant form may not be the only destination of the conformational conversion. It may be the reason why ChPrP was more resistant to PK alone than ChPrP with Hep. Sorry for not including the acrylamide concentration in the MM section. The acrylamide concentration (12%) was used in this study. Details were added to the revised manuscript at Page 6 Lines 109-111 and Page 7 Lines 115-116.

- The authors used CD spectroscopy to show changes of PrP secondary structure. “When MoPrP was treated with 30 μg/mL heparin, a dramatic decrease in the amplitude of the CD spectrum is observed (Fig 5a), the associated β-sheet content of which was 13.8-fold increase compared to MoPrP in solution. In contrast, incubating of the ChPrP to 30 μg/mL heparin induces a relatively small decrease in the amplitude of the CD spectrum (Fig 5b), and only 1.3-fold increase in the β-sheet content was estimated relative to the secondary structure of ChPrP in solution” (lines 199-205).

The authors did not describe how they calculated the beta sheet content, and whether they used any spectrum deconvolution. A visual analysis of the result is not sufficient for this result. The data obtained with ChPrP + Hep 10 was not described. In addition, the MoPrP + Hep 30 data does not show enough ellipticity signal to carry out deconvolution. As the authors point out, the CD is an absorptive technique and changes in sample turbidity impair the analysis of secondary structure content. Another technique like FTIR would be more appropriate in this case. The dichroism data seems to corroborate with a greater aggregation and, therefore, a greater loss of signal, as observed in the TEM data of figure 6, but this does not allow inference about changes in the protein secondary structure. However, the data shows a different effect between murine and chicken proteins.

Response: Thank you for your good suggestions. The β-sheet content was calculated by the software of the chiascan spectropolarimeter (Applied photophysics). It is really true as the reviewer suggested that the MoPrP + Hep 30 data does not show enough ellipticity signal, thus we deleted the calculated β-sheet content of MoPrP+ Hep 30. Details could be found in the revised manuscript at Page 25 Lines 491-494. The data obtained with ChPrP + Hep 10 was described in the revised manuscript at Page 13 Lines 245-247. It is really true as the reviewer suggested that FTIR would be more appropriate in this case, whereas this instrument is not available in our university at present. If there is an opportunity in the future, we will definitely consider using FTIR for investigating secondary structures. In this study, it was intended to compare the different effects of heparin on ChPrP and MoPrP. Although changes in sample turbidity impair the analysis of secondary structure content using CD, different effect of 30 μg/mL heparin on murine and chicken proteins was observed.

- “As for ChPrP treated with 30 μg/mL heparin, less PK resistance and slight increase of β-sheet structure was observed, suggesting that no significant conformational changes occurred and the infectious form PrPSc may not be the only destination of the conformational conversion [47, 48].” (lines 267-270). Conclusions like this must be reconsidered and rephrased.

Responses: Thanks for your suggestion. We revised the related sentences to delete the inappropriate speculation. The revised parts can be found in the revised manuscript at Page 17 Lines 329-330.

Minor comments:

- Running title: I suggest changing for a more specific running title.

Responses: Thanks for your suggestion. We have changed the running title to “Effect of heparin on ChPrP and MoPrP aggregation”.

- All data were obtained in water or sodium acetate buffer, with no NaCl. The interaction of Hep with PrP is electrostatic, modulated by salt concentration. No NaCl may enhance the aggregation induced by Hep. Absence of NaCl can also mask possible differences between affinity for murine and chicken PrP. I suggest performing the experiment over different NaCl concentrations, comparing with the literature. Indeed, the effect showed to be different between the two PrP constructs.

Responses: Thanks for your good suggestion. In our current study, Thioflavin T fluorescence of PrP was measured both in MilliQ water and sodium acetate buffer. The results were consistent in both solutions. To avoid the impact of buffers, subsequent experiments are carried out in the MilliQ water. Indeed, the effect was showed to be different between the two PrP constructs. Thanks very much for the reviewer’s good suggestion, which gave us more inspiration to study the differences between affinity of murine and chicken PrP to cofactors, which would be carried out in detail in our future studies.

- The authors centrifuged PrP:Hep samples and analysed the pellet at Figure 3. In the text its written “almost all PrPs treated with 10 μg/mL heparin were found in the pellet fraction after centrifugation (data not shown)” (line 180 and 181). I suggest showing the data, not only to compare the effect of different concentrations of Hep, but also to show how much of the total protein aggregates.

Responses: Thanks for your good suggestion. We have revised the Fig 3 to support “almost all PrPs treated with 10 μg/mL heparin were found in the pellet fraction after centrifugation” and deleted “data not shown” in the revised manuscript on Page 12 Line 219. The newly revised Fig 3 was as followings:

- MoPrP and ChPrP seems to be aggregated before Hep addition. Authors should consider this when analyzing all the data and interpreting the effect of heparin.

Responses: Thanks for your good suggestion. Indeed, a small amount of MoPrP and ChPrP precipitated even in the absence of heparin (Fig 3). Therefore, high speed centrifugation (14,000 g, 4°C, 30 min) was performed before all experiments in this manuscript. Moreover, PrP alone sample, regarded as a control sample, was included in all experiments and the effect of heparin was summarized through comparative analysis of the characteristics of PrP in the absence or presence of heparin.

- Hep induced the formation of PrP spherical aggregates in reference 39 (supporting material), similar to what is being showed. The author should cite and compare.

Responses: Thanks for your good advices and for suggesting the important reference. We have added the comparison of PrP spherical aggregates to the revised manuscript at Page 14 Lines 265-269. The added sentences were as followings: These spherical aggregates were smaller than those reported in [38], with even more than 200 nm in size. This different was possibly due to using different heparin (30 μg/mL heparin v.s. 2 μM low molecular weight heparin) and incubating in different buffers (MilliQ water v.s. 10 mM acetate and 100 mM NaCl, pH 5.5).

- “Our result showed that heparin binding didn’t play a role in amyloid fibril formation under denaturation (Fig 1). However, low-molecular-weight heparin was shown to increase the stability of full-length PrPC from mice under denaturation and seeded with infected brain tissue homogenate [43]” (lines 235-238). The authors should consider differences on sample preparation: PrP:Hep interaction was in denaturing buffer (this paper) and before addition to denaturing buffer (reference 43); Hep concentration and buffer are different.

Responses: Thanks for your good advices and for suggesting the important reference. In the reference 43, LMWHep (final concentration, 25 μM) was added to the denaturing buffer containing 0.1 mg/mL rPrP (similar to this paper), or previously to the rPrP solution before rPrP was added to the denaturing buffer. The addition of LMWHep to denaturing buffer containing 0.1 mg/mL rPrP (similar to this paper) caused a delay in rPrP fibril formation when the reaction was seeded with infected mouse BH. However, our result showed that heparin addition didn’t play a role in amyloid fibril formation under denaturing conditions. This different was possibly due to using different heparin (30 μg/mL heparin v.s. 25 μM low molecular weight heparin) and incubating in different buffers (100 mM potassium phosphate buffer, 2 mol/L GuHCl, pH 6.5 v.s. 10 mM phosphate buffer, 130 mM NaCl, pH 7.4). Thus, we revised the related sentences in the revised manuscript Page 15 Lines 299-302. 

- “The present study confirmed that heparin can decrease MoPrP and ChPrP stability and facilitate them to aggregate. Furthermore, our study revealed that biochemical properties of the aggregates differed depending on heparin concentrations.” (lines 247-250). The authors should be careful with this statement since inducing a decrease in solubility is not necessarily linked to a decrease in stability of the monomeric protein. Therefore, other experiments would be necessary to evaluate changes in stability.

Responses: Thanks for your good suggestions. We have revised the related sentence as suggested by reviewer. More details had been added to the revised manuscript at Page 7 Lines 124-127. The revised parts were as followings: The present study confirmed that heparin can decrease MoPrP and ChPrP solubility and facilitate them to aggregate. Furthermore, our study revealed that biochemical properties of the aggregates differed depending on heparin concentrations.

Thank you very much indeed for your time and efforts to help us improve the quality of this MS.

We hope that the reviewers and the editors are satisfactory with the above responses.

Once again, many thanks to the reviewers and the editor.

---

## [Decision Letter · Decision Letter 1]

15 Jan 2021

PONE-D-20-18047R1

Comparative analysis of heparin affecting the biochemical properties of chicken and murine prion proteins

PLOS ONE

Dear Dr. Wang,

Thank you for submitting your manuscript to PLOS ONE. After careful consideration, we feel that it has merit but does not fully meet PLOS ONE’s publication criteria as it currently stands. One of the reviewers still has major concerns about your manuscript that I would ask you to address if possible in a second revision.

We look forward to receiving your revised manuscript.

Kind regards,

Byron Caughey

Academic Editor

PLOS ONE

Reviewers' comments:

Reviewer's Responses to Questions

**Comments to the Author**

1. If the authors have adequately addressed your comments raised in a previous round of review and you feel that this manuscript is now acceptable for publication, you may indicate that here to bypass the “Comments to the Author” section, enter your conflict of interest statement in the “Confidential to Editor” section, and submit your "Accept" recommendation.

Reviewer #1: All comments have been addressed

Reviewer #2: (No Response)

2. Is the manuscript technically sound, and do the data support the conclusions?

Reviewer #1: Yes

Reviewer #2: Partly

3. Has the statistical analysis been performed appropriately and rigorously? 

Reviewer #1: Yes

Reviewer #2: N/A

4. Have the authors made all data underlying the findings in their manuscript fully available?

Reviewer #1: Yes

Reviewer #2: Yes

5. Is the manuscript presented in an intelligible fashion and written in standard English?

Reviewer #1: Yes

Reviewer #2: Yes

6. Review Comments to the Author

Reviewer #1: I am satisfied with the improvements made by the authors to this manuscript. I acknowledge their hard work and look forward to their future studies.

Reviewer #2: Dear Authors,

I still have important criticisms about the data and interpretations presented in the paper “Comparative analysis of heparin affecting the biochemical properties of chicken and murine prion proteins”, and for this reason, I forward my observations for your appreciation. Here are the first observations made by me, with the authors' response and my counterarguments.

1 - The experiments in figure 2 used ThT to follow PrP aggregation. However, the fiber morphology by TEM, and the CD signal are not characteristic of fibrillar forms. PrP:Hep 30 ug/mL samples showed the highest ThT fluorescence, but the smallest pellet. The authors should address and explain this contradiction. I suggest checking if the light scattering signal is not interfering with ThT signal. The authors should also show the ThT spectra of the samples with great variation to discard this doubt. The author could also use other approaches to confirm fibril formation.

Author Response: Thank you for your good suggestions. In figure 1, we studied the effects of heparin on fibril formation of PrP and found that heparin may not play a crucial part in amyloid fibril formation under denaturing conditions. In the rest of the manuscript, we did not study the formation and characteristics of fibers. The CD and TEM were used to monitor the conformational changes and aggregates of PrP. Indeed, PrP:Hep 30 μg/mL samples showed the highest ThT fluorescence, but the smallest pellet. However, the protein in the pellet did not have to be aggregated and electron micrographs of resuspended pellet showed no significant aggregates. Protein aggregation into amyloid fibrils is a pathological hall-mark of many diseases, including prion. We would investigate fibril formation in detail in our future studies. Thanks very much for the reviewer’s good suggestion again.

Reviewer Response: although the authors answered that they evaluated the formation of fibers only at Fig1, they used ThT in Fig2 and observed the morphology of aggregates in Fig6. There are serious inconsistencies between the results observed in Fig 2, 3, 5 and 6. The authors argue that although showing the highest ThT signal, the sample in the presence of 30 ug/mL of Heparin forms a smaller pellet (Fig3), because “protein in the pellet did not have to be aggregated and electron micrographs of resuspended pellet showed no significant aggregates".

The first problem of this answer is that the protein being studied will only form a pellet after centrifugation for 10 min at 14000 g (line 119) if it is in an oligomeric/aggregated state. The second problem of this answer is that Fig 6 shows significant aggregates for MoPrP+Hep30. The aggregate state is not directly related to the fiber morphology, not having a direct relationship with the observed signal. However, since there is ThT fluorescence and pellet, the authors should observe the morphology of protofibers and/or fibers in this electron micrographs. Author should explain this contradiction or show that the sample has soluble fibers that did not go to the pellet.

The authors reported that the PrP+Hep10 sample was almost entirely going to the pellet (although they have not yet shown the relationship of how much remained in the supernatant at this and the other conditions studied) (Fig3). However, this PrP:Hep sample showed the greatest sign of ellipticity (Fig 5) and was less aggregated in the electron micrograph (Fig 6), suggesting its less aggregated. The authors should discuss this.

2 - PK treatment was performed in order to show if the aggregates formed were resistant, “Resistance to digestion with PK is one of the characteristics of PrP fibrils and PrPSc” (line 178 and 179).

Figure 4 shows this data, but figures are not of good quality as protein is degraded by PK at concentrations lower than usually used for PrP fibrils or PrPSc. PrP:Hep 30 ug/mL was used, although it was less aggregated than PrP:Hep 10 ug/mL (Figure 3). ChPrP was more resistant to PK alone than MoPrP and both samples with Hep, although ChPrP:Hep 30 ug/mL showed the highest ThT fluorescence (Figure 1). Authors must clarify and discuss these contradictions. Also, authors should include the acrylamide concentration in the MM section.

Author Response: Thank you for your good suggestions. In Fig 4, PrP was incubated with heparin in MilliQ water without any seeds, which was not a suitable condition for fibril formation. Under this condition, the interaction between PrP and cofactor causes widespread and persistent conformational changes to form an intermediate species of PrPSc (A. Zurawel, et al. Biochemistry, 2014, doi: 10.1021/bi4014825). These may be the reason why PK concentrations used in Fig 4 were lower than usually used for PrP fibrils or PrPSc. Indeed, PrP incubated with low concentration of heparin (10 μg/mL) were less soluble than those treated with heparin at high concentration (30 μg/mL) (Fig 3). However, the protein in the pellet did not have to be aggregated. The presence of pellet did not imply conformational changes, and most of the protein in the pellet is not conducive to subsequent experiments. Thus, 30 μg/mL heparin was used when we analyzed the PK resistance. ChPrP:Hep 30 μg/mL showed the highest ThT fluorescence (Fig 1), which suggests conformational changes were occurred after incubation with heparin. However, PK-resistant form may not be the only destination of the conformational conversion. It may be the reason why ChPrP was more resistant to PK alone than ChPrP with Hep. Sorry for not including the acrylamide concentration in the MM section. The acrylamide concentration (12%) was used in this study. Details were added to the revised manuscript at Page 6 Lines 109-111 and Page 7 Lines 115-116.

Reviewer Response: When the authors answer “These may be the reason why PK concentrations used in Fig 4 were lower than usually used for PrP fibrils or PrPSc” they should discuss it in the paper for readers. The authors answered “However, the protein in the pellet did not have to be aggregated. The presence of pellet did not imply conformational changes, and most of the protein in the pellet is not conducive to subsequent experiments”. Again, over the centrifugation performed, the presence of the monomeric form of the protein forming the pellet would be very unlikely, which would have to be shown, although the authors are right when they say that there is not necessarily a conformational change, which would have to be shown as well. But once there is a ThT fluorescence increment (Fig 1), there is a suggestion of conformational change.

I agree when the authors wrote “However, PK-resistant form may not be the only destination of the conformational conversion” and they should discuss this for readers. But, when they wrote “It may be the reason why ChPrP was more resistant to PK alone than ChPrP with Hep”, this argument is very unconvincing.

3 - The authors used CD spectroscopy to show changes of PrP secondary structure. “When MoPrP was treated with 30 μg/mL heparin, a dramatic decrease in the amplitude of the CD spectrum is observed (Fig 5a), the associated β-sheet content of which was 13.8-fold increase compared to MoPrP in solution. In contrast, incubating of the ChPrP to 30 μg/mL heparin induces a relatively small decrease in the amplitude of the CD spectrum (Fig 5b), and only 1.3-fold increase in the β-sheet content was estimated relative to the secondary structure of ChPrP in solution” (lines 199-205).

The authors did not describe how they calculated the beta sheet content, and whether they used any spectrum deconvolution. A visual analysis of the result is not sufficient for this result. The data obtained with ChPrP + Hep 10 was not described. In addition, the MoPrP + Hep 30 data does not show enough ellipticity signal to carry out deconvolution. As the authors point out, the CD is an absorptive technique and changes in sample turbidity impair the analysis of secondary structure content. Another technique like FTIR would be more appropriate in this case. The dichroism data seems to corroborate with a greater aggregation and, therefore, a greater loss of signal, as observed in the TEM data of figure 6, but this does not allow inference about changes in the protein secondary structure. However, the data shows a different effect between murine and chicken proteins.

Author Response: Thank you for your good suggestions. The β-sheet content was calculated by the software of the chiascan spectropolarimeter (Applied photophysics). It is really true as the reviewer suggested that the MoPrP + Hep 30 data does not show enough ellipticity signal, thus we deleted the calculated β-sheet content of MoPrP+ Hep 30. Details could be found in the revised manuscript at Page 25 Lines 491-494. The data obtained with ChPrP + Hep 10 was described in the revised manuscript at Page 13 Lines 245-247. It is really true as the reviewer suggested that FTIR would be more appropriate in this case, whereas this instrument is not available in our university at present. If there is an opportunity in the future, we will definitely consider using FTIR for investigating secondary structures. In this study, it was intended to compare the different effects of heparin on ChPrP and MoPrP. Although changes in sample turbidity impair the analysis of secondary structure content using CD, different effect of 30 μg/mL heparin on murine and chicken proteins was observed.

Reviewer Response: The last version of the manuscript still has sentences about calculated beta sheet content as “In contrast, incubating of the ChPrP to 30 μg/mL heparin induces a relatively small decrease in the amplitude of the CD spectrum (Fig 5b), and only 1.3-fold increase in the β-sheet content was estimated relative to the secondary structure of ChPrP in solution.” The previous sentence says “When MoPrP was treated with 30 μg/mL heparin, a dramatic decrease in the amplitude of the CD spectrum is observed (Fig 5a)”, so they are describing different effects for the same sample.

4 - “As for ChPrP treated with 30 μg/mL heparin, less PK resistance and slight increase of β-sheet structure was observed, suggesting that no significant conformational changes occurred and the infectious form PrPSc may not be the only destination of the conformational conversion [47, 48].” (lines 267-270). Conclusions like this must be reconsidered and rephrased.

Responses: Thanks for your suggestion. We revised the related sentences to delete the inappropriate speculation. The revised parts can be found in the revised manuscript at Page 17 Lines 329-330.

Reviewer Response: ok

Minor comments:

1- Running title: I suggest changing for a more specific running title.

Author Response: Thanks for your suggestion. We have changed the running title to “Effect of heparin on ChPrP and MoPrP aggregation”.

Reviewer Response: ok

2 - All data were obtained in water or sodium acetate buffer, with no NaCl. The interaction of Hep with PrP is electrostatic, modulated by salt concentration. No NaCl may enhance the aggregation induced by Hep. Absence of NaCl can also mask possible differences between affinity for murine and chicken PrP. I suggest performing the experiment over different NaCl concentrations, comparing with the literature. Indeed, the effect showed to be different between the two PrP constructs.

Responses: Thanks for your good suggestion. In our current study, Thioflavin T fluorescence of PrP was measured both in MilliQ water and sodium acetate buffer. The results were consistent in both solutions. To avoid the impact of buffers, subsequent experiments are carried out in the MilliQ water. Indeed, the effect was showed to be different between the two PrP constructs. Thanks very much for the reviewer’s good suggestion, which gave us more inspiration to study the differences between affinity of murine and chicken PrP to cofactors, which would be carried out in detail in our future studies.

Reviewer Response: Author should include this perspective and its importance in discussion for the readers.

3 - The authors centrifuged PrP:Hep samples and analysed the pellet at Figure 3. In the text its written “almost all PrPs treated with 10 μg/mL heparin were found in the pellet fraction after centrifugation (data not shown)” (line 180 and 181). I suggest showing the data, not only to compare the effect of different concentrations of Hep, but also to show how much of the total protein aggregates.

Responses: Thanks for your good suggestion. We have revised the Fig 3 to support “almost all PrPs treated with 10 μg/mL heparin were found in the pellet fraction after centrifugation” and deleted “data not shown” in the revised manuscript on Page 12 Line 219. The newly revised Fig 3 was as followings:

Reviewer Response: when calculating the fraction present in the pellet of PrP:Hep samples using the densitometry of the band referring to the pellet of PrP alone (which must have negligible amounts of protein in the pellet), the authors do not give us the real dimension of how many of the 5 µM of protein was going to the pellet. This does not invalidate the observation made; however, it would be very enlightening if we had information on how much of the total protein is going into the pellet.

4 - MoPrP and ChPrP seems to be aggregated before Hep addition. Authors should consider this when analyzing all the data and interpreting the effect of heparin.

Responses: Thanks for your good suggestion. Indeed, a small amount of MoPrP and ChPrP precipitated even in the absence of heparin (Fig 3). Therefore, high speed centrifugation (14,000 g, 4°C, 30 min) was performed before all experiments in this manuscript. Moreover, PrP alone sample, regarded as a control sample, was included in all experiments and the effect of heparin was summarized through comparative analysis of the characteristics of PrP in the absence or presence of heparin.

Reviewer Response: Authors included the sentence “The pellet was separated by centrifugation (14,000 g, 30 min) and the supernatant was used for further analysis” in Circular dichroism (CD) and transmission electron microscopy (TEM) subsection of MM. As it is written, it is giving the impression to the reader that after incubation with the ligand, the protein went through this centrifugation step and the supernatant was analyzed by CD and TEM, but the results should show analysis of the pellet.

Fig 3 and 6 show that PrP alone, even after the centrifugation performed, shows some aggregates. As the authors answered, this sample was correctly used as a control in all experiments. Even so, the authors should consider in the discussion a possible effect of the ligand over the balance between aggregated and non-aggregated species.

5 - Hep induced the formation of PrP spherical aggregates in reference 39 (supporting material), similar to what is being showed. The author should cite and compare.

Responses: Thanks for your good advices and for suggesting the important reference. We have added the comparison of PrP spherical aggregates to the revised manuscript at Page 14 Lines 265-269. The added sentences were as followings: These spherical aggregates were smaller than those reported in [38], with even more than 200 nm in size. This different was possibly due to using different heparin (30 μg/mL heparin v.s. 2 μM low molecular weight heparin) and incubating in different buffers (MilliQ water v.s. 10 mM acetate and 100 mM NaCl, pH 5.5).

Reviewer Response: ok. Review grammar of the new sentence.

6 - “Our result showed that heparin binding didn’t play a role in amyloid fibril formation under denaturation (Fig 1). However, low-molecular-weight heparin was shown to increase the stability of full-length PrPC from mice under denaturation and seeded with infected brain tissue homogenate [43]” (lines 235-238). The authors should consider differences on sample preparation: PrP:Hep interaction was in denaturing buffer (this paper) and before addition to denaturing buffer (reference 43); Hep concentration and buffer are different.

Responses: Thanks for your good advices and for suggesting the important reference. In the reference 43, LMWHep (final concentration, 25 μM) was added to the denaturing buffer containing 0.1 mg/mL rPrP (similar to this paper), or previously to the rPrP solution before rPrP was added to the denaturing buffer. The addition of LMWHep to denaturing buffer containing 0.1 mg/mL rPrP (similar to this paper) caused a delay in rPrP fibril formation when the reaction was seeded with infected mouse BH. However, our result showed that heparin addition didn’t play a role in amyloid fibril formation under denaturing conditions. This different was possibly due to using different heparin (30 μg/mL heparin v.s. 25 μM low molecular weight heparin) and incubating in different buffers (100 mM potassium phosphate buffer, 2 mol/L GuHCl, pH 6.5 v.s. 10 mM phosphate buffer, 130 mM NaCl, pH 7.4). Thus, we revised the related sentences in the revised manuscript Page 15 Lines 299-302.

Reviewer Response: ok

7 - “The present study confirmed that heparin can decrease MoPrP and ChPrP stability and facilitate them to aggregate. Furthermore, our study revealed that biochemical properties of the aggregates differed depending on heparin concentrations.” (lines 247-250). The authors should be careful with this statement since inducing a decrease in solubility is not necessarily linked to a decrease in stability of the monomeric protein. Therefore, other experiments would be necessary to evaluate changes in stability.

Responses: Thanks for your good suggestions. We have revised the related sentence as suggested by reviewer. More details had been added to the revised manuscript at Page 7 Lines 124-127. The revised parts were as followings: The present study confirmed that heparin can decrease MoPrP and ChPrP solubility and facilitate them to aggregate. Furthermore, our study revealed that biochemical properties of the aggregates differed depending on heparin concentrations.

Reviewer Response: ok

7. PLOS authors have the option to publish the peer review history of their article (what does this mean?). If published, this will include your full peer review and any attached files.

Reviewer #1: **Yes: **Laura J Ellett

Reviewer #2: **Yes: **Tuane C R G Vieira

---

## [Author Response · Author response to Decision Letter 1]

26 Jan 2021

Dear Byron Caughey and Dear Reviewers:

Thank you for your efforts to handle and review our revised manuscript (PONE-D-20-18047_R1) entitled: “Comparative analysis of heparin affecting the biochemical properties of chicken and murine prion proteins”. We have carefully revised the manuscript based on your suggestions. Followings are Listed Responses to the Editor and Reviewers’ Comments and Suggestions:

Thank you for submitting your manuscript to PLOS ONE. After careful consideration, we feel that it has merit but does not fully meet PLOS ONE’s publication criteria as it currently stands. One of the reviewers still has major concerns about your manuscript that I would ask you to address if possible in a second revision.

Response: We have addressed the reviewers' comments carefully below. We appreciate the Editor for the opportunity.

Reviewer #1: I am satisfied with the improvements made by the authors to this manuscript. I acknowledge their hard work and look forward to their future studies.

Response: Thank you for your positive comments.

Reviewer #2: Dear Authors,

I still have important criticisms about the data and interpretations presented in the paper “Comparative analysis of heparin affecting the biochemical properties of chicken and murine prion proteins”, and for this reason, I forward my observations for your appreciation. Here are the first observations made by me, with the authors' response and my counterarguments.

Response: Thank you for your comments. We appreciate the reviewer’s efforts to read this paper and previous paper. We have addressed the review’s concerns point by point below.

1 - The experiments in figure 2 used ThT to follow PrP aggregation. However, the fiber morphology by TEM, and the CD signal are not characteristic of fibrillar forms. PrP:Hep 30 ug/mL samples showed the highest ThT fluorescence, but the smallest pellet. The authors should address and explain this contradiction. I suggest checking if the light scattering signal is not interfering with ThT signal. The authors should also show the ThT spectra of the samples with great variation to discard this doubt. The author could also use other approaches to confirm fibril formation.

Author Response: Thank you for your good suggestions. In figure 1, we studied the effects of heparin on fibril formation of PrP and found that heparin may not play a crucial part in amyloid fibril formation under denaturing conditions. In the rest of the manuscript, we did not study the formation and characteristics of fibers. The CD and TEM were used to monitor the conformational changes and aggregates of PrP. Indeed, PrP:Hep 30 μg/mL samples showed the highest ThT fluorescence, but the smallest pellet. However, the protein in the pellet did not have to be aggregated and electron micrographs of resuspended pellet showed no significant aggregates. Protein aggregation into amyloid fibrils is a pathological hall-mark of many diseases, including prion. We would investigate fibril formation in detail in our future studies. Thanks very much for the reviewer’s good suggestion again.

Reviewer Response: although the authors answered that they evaluated the formation of fibers only at Fig1, they used ThT in Fig2 and observed the morphology of aggregates in Fig6. There are serious inconsistencies between the results observed in Fig 2, 3, 5 and 6. The authors argue that although showing the highest ThT signal, the sample in the presence of 30 ug/mL of Heparin forms a smaller pellet (Fig3), because “protein in the pellet did not have to be aggregated and electron micrographs of resuspended pellet showed no significant aggregates".

The first problem of this answer is that the protein being studied will only form a pellet after centrifugation for 10 min at 14000 g (line 119) if it is in an oligomeric/aggregated state.

Response: Thanks for your suggestions. It is really true as the reviewer suggested that the protein being studied will only form a pellet after centrifugation if it is in an oligomeric/aggregated state. Sorry for the confuse. What I mean is the protein in the pellet did not have to undergo a conformational change. PrP:Hep 30 μg/mL samples showed the highest ThT fluorescence, but the smallest pellet. In contrast, PrP:Hep 10 μg/mL samples showed lower ThT fluorescence, but the more pellet. Such results have previously been reported (Ellett LJ, et al, 2015, Glycobiology. Doi: 10.1093/glycob/cwv014.). HS treated recMoPrP showed visible turbidity, whereas the ThT fluorescence of those were low. The contradiction might be explained by that the protein in the pellet did not have to undergo a conformational change.

The second problem of this answer is that Fig 6 shows significant aggregates for MoPrP+Hep30. The aggregate state is not directly related to the fiber morphology, not having a direct relationship with the observed signal. However, since there is ThT fluorescence and pellet, the authors should observe the morphology of protofibers and/or fibers in this electron micrographs. Author should explain this contradiction or show that the sample has soluble fibers that did not go to the pellet.

Response: Thanks for your good suggestions. Indeed, MoPrP+Hep30 showed high highest ThT fluorescence. The increase of ThT fluorescence suggests a conformational change, but does not necessarily imply fiber formation. In addition, the experiments of Fig 2 to Fig 6 were all performed under physiological conditions, whereas amyloid fibrils were usually formed under denaturing conditions. Thus, we didn’t observe the morphology of protofibers and/or fibers in electron micrographs. Similar results were observed in Ellett LJ, et al, 2015, Glycobiology. Doi: 10.1093/glycob/cwv014. Only aggregates were observed when recMoPrP was incubated with GAGs under physiological conditions.

The authors reported that the PrP+Hep10 sample was almost entirely going to the pellet (although they have not yet shown the relationship of how much remained in the supernatant at this and the other conditions studied) (Fig3). However, this PrP:Hep sample showed the greatest sign of ellipticity (Fig 5) and was less aggregated in the electron micrograph (Fig 6), suggesting its less aggregated. The authors should discuss this. 

Response: Thanks for your good suggestions. Pellet fraction of MoPrP+Hep10 and ChPrP+Hep10 samples were approximately 75%~ 80% (Fig 3). The pellet was separated by centrifugation (14,000 g, 30 min) and the supernatant was used for CD analysis. The CD spectra was significantly lost (almost changed to smooth curves) when 10 μg/mL heparin was used (Fig 5), which was consistent with the formation of insoluble aggregates (visible turbidity). Electron micrographs of PrP+Hep10 samples showed no significant aggregates. Electron micrographs of resuspended pellet showed no significant aggregates, either. The protein pellet could not attach well to the copper grids, whereas the oligomeric protein could. This might explain why PrP+Hep10 samples were less aggregated.

2 - PK treatment was performed in order to show if the aggregates formed were resistant, “Resistance to digestion with PK is one of the characteristics of PrP fibrils and PrPSc” (line 178 and 179).

Figure 4 shows this data, but figures are not of good quality as protein is degraded by PK at concentrations lower than usually used for PrP fibrils or PrPSc. PrP:Hep 30 ug/mL was used, although it was less aggregated than PrP:Hep 10 ug/mL (Figure 3). ChPrP was more resistant to PK alone than MoPrP and both samples with Hep, although ChPrP:Hep 30 ug/mL showed the highest ThT fluorescence (Figure 1). Authors must clarify and discuss these contradictions. Also, authors should include the acrylamide concentration in the MM section.

Author Response: Thank you for your good suggestions. In Fig 4, PrP was incubated with heparin in MilliQ water without any seeds, which was not a suitable condition for fibril formation. Under this condition, the interaction between PrP and cofactor causes widespread and persistent conformational changes to form an intermediate species of PrPSc (A. Zurawel, et al. Biochemistry, 2014, doi: 10.1021/bi4014825). These may be the reason why PK concentrations used in Fig 4 were lower than usually used for PrP fibrils or PrPSc. Indeed, PrP incubated with low concentration of heparin (10 μg/mL) were less soluble than those treated with heparin at high concentration (30 μg/mL) (Fig 3). However, the protein in the pellet did not have to be aggregated. The presence of pellet did not imply conformational changes, and most of the protein in the pellet is not conducive to subsequent experiments. Thus, 30 μg/mL heparin was used when we analyzed the PK resistance. ChPrP:Hep 30 μg/mL showed the highest ThT fluorescence (Fig 1), which suggests conformational changes were occurred after incubation with heparin. However, PK-resistant form may not be the only destination of the conformational conversion. It may be the reason why ChPrP was more resistant to PK alone than ChPrP with Hep. Sorry for not including the acrylamide concentration in the MM section. The acrylamide concentration (12%) was used in this study. Details were added to the revised manuscript at Page 6 Lines 109-111 and Page 7 Lines 115-116.

Reviewer Response: When the authors answer “These may be the reason why PK concentrations used in Fig 4 were lower than usually used for PrP fibrils or PrPSc” they should discuss it in the paper for readers. The authors answered “However, the protein in the pellet did not have to be aggregated. The presence of pellet did not imply conformational changes, and most of the protein in the pellet is not conducive to subsequent experiments”. Again, over the centrifugation performed, the presence of the monomeric form of the protein forming the pellet would be very unlikely, which would have to be shown, although the authors are right when they say that there is not necessarily a conformational change, which would have to be shown as well. But once there is a ThT fluorescence increment (Fig 1), there is a suggestion of conformational change.

I agree when the authors wrote “However, PK-resistant form may not be the only destination of the conformational conversion” and they should discuss this for readers. But, when they wrote “It may be the reason why ChPrP was more resistant to PK alone than ChPrP with Hep”, this argument is very unconvincing.

Response: Thanks for your good suggestions. We have added additional discussion on why PK concentrations used in Fig 4 were lower than usually used for PrP fibrils or PrPSc. More details had been added to the revised manuscript at Page 17 Lines 326-332. The revised parts were as followings: Heparin-treated MoPrP was more PK resistant, whereas the MoPrP was degraded by PK at concentrations lower than usually used for PrP fibrils or PrPSc. Here, PrP was incubated with heparin in MilliQ water without any seeds, which was not a suitable condition for fibril formation. Under this condition, the interaction between PrP and heparin causes widespread and persistent conformational changes to form an intermediate species of PrPSc [46]. These may be the reason why PK concentrations used here were lower than usually used for PrP fibrils or PrPSc.

Sorry for the confuse. After incubating of PrP and heparin, centrifugation was performed only before CD, whereas other experiments, including TEM, were performed without centrifugation. To avoid confusion, we revised the description of TEM method. More details had been added to the revised manuscript at Page 7 Lines 122-127. The revised parts were as followings: The pellet was separated by centrifugation (14,000 g, 30 min) and the supernatant was used for CD analysis. The ellipticity values (MilliQ water or heparin solution) were used as controls. For TEM, 4 μL incubating solution not centrifuged was fixed on 300 mesh copper grids (BZ10023b, Zhongjingkeyi), washed with 4 μL water, negatively stained using 2% uranyl acetate and examined on a Tecnai G2 Spirit TEM at voltage of 120 kV.

It is really true as the reviewer suggested that “once there is a ThT fluorescence increment (Fig 1), there is a suggestion of conformational change.” Protein conformational changes might be masked by aggregates in CD. FTIR would be more appropriate in this case, whereas this instrument is not available in our university at present. If there is an opportunity in the future, we will definitely consider using FTIR for investigating secondary structures.

We have added additional discussion and deleted the unconvincing sentences as the reviewer suggested. More details had been added to the revised manuscript at Page 17 Lines 332-338. The revised parts were as followings: In contrast, ChPrP was more resistant to PK alone than ChPrP with 30 μg/mL heparin. The ChPrP treated with 30 μg/mL heparin showed the highest ThT fluorescence (Fig 2b), which suggests conformational changes were occurred after incubation with heparin. The formation of β-sheet conformation in prion protein usually leads to a protease-resistant form [19,20]. However, PK-resistant form may not be the only destination of the conformational conversion [47].

3 - The authors used CD spectroscopy to show changes of PrP secondary structure. “When MoPrP was treated with 30 μg/mL heparin, a dramatic decrease in the amplitude of the CD spectrum is observed (Fig 5a), the associated β-sheet content of which was 13.8-fold increase compared to MoPrP in solution. In contrast, incubating of the ChPrP to 30 μg/mL heparin induces a relatively small decrease in the amplitude of the CD spectrum (Fig 5b), and only 1.3-fold increase in the β-sheet content was estimated relative to the secondary structure of ChPrP in solution” (lines 199-205).

The authors did not describe how they calculated the beta sheet content, and whether they used any spectrum deconvolution. A visual analysis of the result is not sufficient for this result. The data obtained with ChPrP + Hep 10 was not described. In addition, the MoPrP + Hep 30 data does not show enough ellipticity signal to carry out deconvolution. As the authors point out, the CD is an absorptive technique and changes in sample turbidity impair the analysis of secondary structure content. Another technique like FTIR would be more appropriate in this case. The dichroism data seems to corroborate with a greater aggregation and, therefore, a greater loss of signal, as observed in the TEM data of figure 6, but this does not allow inference about changes in the protein secondary structure. However, the data shows a different effect between murine and chicken proteins.

Author Response: Thank you for your good suggestions. The β-sheet content was calculated by the software of the chiascan spectropolarimeter (Applied photophysics). It is really true as the reviewer suggested that the MoPrP + Hep 30 data does not show enough ellipticity signal, thus we deleted the calculated β-sheet content of MoPrP+ Hep 30. Details could be found in the revised manuscript at Page 25 Lines 491-494. The data obtained with ChPrP + Hep 10 was described in the revised manuscript at Page 13 Lines 245-247. It is really true as the reviewer suggested that FTIR would be more appropriate in this case, whereas this instrument is not available in our university at present. If there is an opportunity in the future, we will definitely consider using FTIR for investigating secondary structures. In this study, it was intended to compare the different effects of heparin on ChPrP and MoPrP. Although changes in sample turbidity impair the analysis of secondary structure content using CD, different effect of 30 μg/mL heparin on murine and chicken proteins was observed.

Reviewer Response: The last version of the manuscript still has sentences about calculated beta sheet content as “In contrast, incubating of the ChPrP to 30 μg/mL heparin induces a relatively small decrease in the amplitude of the CD spectrum (Fig 5b), and only 1.3-fold increase in the β-sheet content was estimated relative to the secondary structure of ChPrP in solution.” The previous sentence says “When MoPrP was treated with 30 μg/mL heparin, a dramatic decrease in the amplitude of the CD spectrum is observed (Fig 5a)”, so they are describing different effects for the same sample.

Response: Thanks for your good suggestions. We have deleted the sentences about calculated beta sheet content. And we described the effects of 30 μg/mL heparin on different samples (MoPrP and ChPrP) in our last version of the manuscript. Fig 5a and Fig 5b show the CD spectra of MoPrP (a) and ChPrP (b), respectively.

4 - “As for ChPrP treated with 30 μg/mL heparin, less PK resistance and slight increase of β-sheet structure was observed, suggesting that no significant conformational changes occurred and the infectious form PrPSc may not be the only destination of the conformational conversion [47, 48].” (lines 267-270). Conclusions like this must be reconsidered and rephrased.

Responses: Thanks for your suggestion. We revised the related sentences to delete the inappropriate speculation. The revised parts can be found in the revised manuscript at Page 17 Lines 329-330.

Reviewer Response: ok

Response: Thank you for your positive comments.

Minor comments:

1- Running title: I suggest changing for a more specific running title.

Author Response: Thanks for your suggestion. We have changed the running title to “Effect of heparin on ChPrP and MoPrP aggregation”.

Reviewer Response: ok

Response: Thank you for your positive comments.

2 - All data were obtained in water or sodium acetate buffer, with no NaCl. The interaction of Hep with PrP is electrostatic, modulated by salt concentration. No NaCl may enhance the aggregation induced by Hep. Absence of NaCl can also mask possible differences between affinity for murine and chicken PrP. I suggest performing the experiment over different NaCl concentrations, comparing with the literature. Indeed, the effect showed to be different between the two PrP constructs.

Responses: Thanks for your good suggestion. In our current study, Thioflavin T fluorescence of PrP was measured both in MilliQ water and sodium acetate buffer. The results were consistent in both solutions. To avoid the impact of buffers, subsequent experiments are carried out in the MilliQ water. Indeed, the effect was showed to be different between the two PrP constructs. Thanks very much for the reviewer’s good suggestion, which gave us more inspiration to study the differences between affinity of murine and chicken PrP to cofactors, which would be carried out in detail in our future studies.

Reviewer Response: Author should include this perspective and its importance in discussion for the readers. 

Response: Thank you for your good suggestions. We have added additional discussion to the revised manuscript at Page 16 Lines 314-318. The added parts were as followings: Additionally, the electrostatic interaction of heparin with PrP might be modulated by salt concentration. In the current study, the reaction solutions with no NaCl might mask possible differences between affinity for MoPrP and ChPrP, which would be carried out in detail in our future studies.

3 - The authors centrifuged PrP:Hep samples and analysed the pellet at Figure 3. In the text its written “almost all PrPs treated with 10 μg/mL heparin were found in the pellet fraction after centrifugation (data not shown)” (line 180 and 181). I suggest showing the data, not only to compare the effect of different concentrations of Hep, but also to show how much of the total protein aggregates.

Responses: Thanks for your good suggestion. We have revised the Fig 3 to support “almost all PrPs treated with 10 μg/mL heparin were found in the pellet fraction after centrifugation” and deleted “data not shown” in the revised manuscript on Page 12 Line 219. The newly revised Fig 3 was as followings:

Reviewer Response: when calculating the fraction present in the pellet of PrP:Hep samples using the densitometry of the band referring to the pellet of PrP alone (which must have negligible amounts of protein in the pellet), the authors do not give us the real dimension of how many of the 5 µM of protein was going to the pellet. This does not invalidate the observation made; however, it would be very enlightening if we had information on how much of the total protein is going into the pellet.

Response: Thanks for your good suggestions. I am so sorry for the mistake in the legend of revised Fig 3 in our last manuscript. The revised Fig 3 showed the pellet fraction of total PrP, not relative to the pellet of PrP in the absence of heparin. And we revised the legend of Fig 3 at Page 11 Line 211-212. The revised parts were as followings: The fraction of PrP in pellet was quantified using ImageJ software and was shown as a percentage of total protein.

Indeed, small amounts of PrP band were detected in pellet fraction of PrP alone after centrifugation, which was shown in Fig 3 (The “0” represents PrP alone, with no heparin).

4 - MoPrP and ChPrP seems to be aggregated before Hep addition. Authors should consider this when analyzing all the data and interpreting the effect of heparin.

Responses: Thanks for your good suggestion. Indeed, a small amount of MoPrP and ChPrP precipitated even in the absence of heparin (Fig 3). Therefore, high speed centrifugation (14,000 g, 4°C, 30 min) was performed before all experiments in this manuscript. Moreover, PrP alone sample, regarded as a control sample, was included in all experiments and the effect of heparin was summarized through comparative analysis of the characteristics of PrP in the absence or presence of heparin.

Reviewer Response: Authors included the sentence “The pellet was separated by centrifugation (14,000 g, 30 min) and the supernatant was used for further analysis” in Circular dichroism (CD) and transmission electron microscopy (TEM) subsection of MM. As it is written, it is giving the impression to the reader that after incubation with the ligand, the protein went through this centrifugation step and the supernatant was analyzed by CD and TEM, but the results should show analysis of the pellet. 

Fig 3 and 6 show that PrP alone, even after the centrifugation performed, shows some aggregates. As the authors answered, this sample was correctly used as a control in all experiments. Even so, the authors should consider in the discussion a possible effect of the ligand over the balance between aggregated and non-aggregated species.

Response: Thanks for your good suggestions. Sorry for the confuse. After incubating of PrP and heparin, centrifugation was performed only before CD, whereas other experiments, including TEM, were performed without centrifugation. To avoid confusion, we revised the description of TEM method. More details had been added to the revised manuscript at Page 5 Line 9-21. The revised parts were as followings: The pellet was separated by centrifugation (14,000 g, 30 min) and the supernatant was used for CD analysis. The ellipticity values (MilliQ water or heparin solution) were used as controls. For TEM, 4 μL incubating solution not centrifuged was fixed on 300 mesh copper grids (BZ10023b, Zhongjingkeyi), washed with 4 μL water, negatively stained using 2% uranyl acetate and examined on a Tecnai G2 Spirit TEM at voltage of 120 kV.

PrP alone sample, regarded as a control sample, was included in all experiments. The possible effect of heparin over the balance between aggregated and non-aggregated species was summarized through comparative analysis of the characteristics of PrP in the absence or presence of heparin and discussed in the manuscript at Pages 17-18 Lines 338-344 and Lines 350-354. The details were as followings: The presence of 30 μg/mL heparin increased PK resistance and aggregate size of MoPrP (Figs 4 and 6), suggesting that high concentration of heparin induces a conformational change and contribute to the conversion of MoPrP to MoPrPSc. As for ChPrP treated with 30 μg/mL heparin, less PK resistance and slight increase of β-sheet structure was observed. The effects of these cofactors on characteristics of ChPrP differ from those on MoPrP. These results may provide a new perspective on understanding the differences between mammalian and non-mammalian PrP and further on unraveling why prion diseases are only observed in mammals.

5 - Hep induced the formation of PrP spherical aggregates in reference 39 (supporting material), similar to what is being showed. The author should cite and compare.

Responses: Thanks for your good advices and for suggesting the important reference. We have added the comparison of PrP spherical aggregates to the revised manuscript at Page 14 Lines 265-269. The added sentences were as followings: These spherical aggregates were smaller than those reported in [38], with even more than 200 nm in size. This different was possibly due to using different heparin (30 μg/mL heparin v.s. 2 μM low molecular weight heparin) and incubating in different buffers (MilliQ water v.s. 10 mM acetate and 100 mM NaCl, pH 5.5).

Reviewer Response: ok. Review grammar of the new sentence.

Response: Thank you for your positive comments and good suggestions. We have checked all the new sentences in the revised manuscript.

6 - “Our result showed that heparin binding didn’t play a role in amyloid fibril formation under denaturation (Fig 1). However, low-molecular-weight heparin was shown to increase the stability of full-length PrPC from mice under denaturation and seeded with infected brain tissue homogenate [43]” (lines 235-238). The authors should consider differences on sample preparation: PrP:Hep interaction was in denaturing buffer (this paper) and before addition to denaturing buffer (reference 43); Hep concentration and buffer are different.

Responses: Thanks for your good advices and for suggesting the important reference. In the reference 43, LMWHep (final concentration, 25 μM) was added to the denaturing buffer containing 0.1 mg/mL rPrP (similar to this paper), or previously to the rPrP solution before rPrP was added to the denaturing buffer. The addition of LMWHep to denaturing buffer containing 0.1 mg/mL rPrP (similar to this paper) caused a delay in rPrP fibril formation when the reaction was seeded with infected mouse BH. However, our result showed that heparin addition didn’t play a role in amyloid fibril formation under denaturing conditions. This different was possibly due to using different heparin (30 μg/mL heparin v.s. 25 μM low molecular weight heparin) and incubating in different buffers (100 mM potassium phosphate buffer, 2 mol/L GuHCl, pH 6.5 v.s. 10 mM phosphate buffer, 130 mM NaCl, pH 7.4). Thus, we revised the related sentences in the revised manuscript Page 15 Lines 299-302.

Reviewer Response: ok

Response: Thank you for your positive comments.

7 - “The present study confirmed that heparin can decrease MoPrP and ChPrP stability and facilitate them to aggregate. Furthermore, our study revealed that biochemical properties of the aggregates differed depending on heparin concentrations.” (lines 247-250). The authors should be careful with this statement since inducing a decrease in solubility is not necessarily linked to a decrease in stability of the monomeric protein. Therefore, other experiments would be necessary to evaluate changes in stability.

Responses: Thanks for your good suggestions. We have revised the related sentence as suggested by reviewer. More details had been added to the revised manuscript at Page 7 Lines 124-127. The revised parts were as followings: The present study confirmed that heparin can decrease MoPrP and ChPrP solubility and facilitate them to aggregate. Furthermore, our study revealed that biochemical properties of the aggregates differed depending on heparin concentrations.

Reviewer Response: ok

Response: Thank you for your positive comments.

We hope that the reviewers and the editors are satisfactory with the above responses.

Once again, many thanks to the two reviewers and the editor.

---

## [Decision Letter · Decision Letter 2]

4 Feb 2021

Comparative analysis of heparin affecting the biochemical properties of chicken and murine prion proteins

PONE-D-20-18047R2

Dear Dr. Wang,

We’re pleased to inform you that your manuscript has been judged scientifically suitable for publication and will be formally accepted for publication once it meets all outstanding technical requirements.

Kind regards,

Byron Caughey

Academic Editor

PLOS ONE

Additional Editor Comments (optional):

Reviewers' comments:

Reviewer's Responses to Questions

**Comments to the Author**

1. If the authors have adequately addressed your comments raised in a previous round of review and you feel that this manuscript is now acceptable for publication, you may indicate that here to bypass the “Comments to the Author” section, enter your conflict of interest statement in the “Confidential to Editor” section, and submit your "Accept" recommendation.

Reviewer #2: All comments have been addressed

2. Is the manuscript technically sound, and do the data support the conclusions?

Reviewer #2: Yes

3. Has the statistical analysis been performed appropriately and rigorously? 

Reviewer #2: N/A

4. Have the authors made all data underlying the findings in their manuscript fully available?

Reviewer #2: Yes

5. Is the manuscript presented in an intelligible fashion and written in standard English?

Reviewer #2: Yes

6. Review Comments to the Author

Reviewer #2: (No Response)

7. PLOS authors have the option to publish the peer review history of their article (what does this mean?). If published, this will include your full peer review and any attached files.

Reviewer #2: **Yes: **Tuane C R G Vieira

---

## [Editor Report · Acceptance letter]

8 Feb 2021

PONE-D-20-18047R2 

Comparative analysis of heparin affecting the biochemical properties of chicken and murine prion proteins 

Dear Dr. Wang:

I'm pleased to inform you that your manuscript has been deemed suitable for publication in PLOS ONE. Congratulations! Your manuscript is now with our production department. 

Kind regards, 

on behalf of

Dr. Byron Caughey 

Academic Editor

PLOS ONE